# The Role of Continental Mesoscale Convective Systems in Forecast Busts within Global Weather Prediction Systems

**David B. Parsons [1],\*, Samuel P. Lillo [1], Christopher P. Rattray [1], Peter Bechtold [2] , Mark J. Rodwell [2] and Connor M. Bruce [1]**

1 School of Meteorology, University of Oklahoma, Norman, OK 73072, USA; splillo@gmail.com (S.P.L.); chris.rattray@ou.edu (C.P.R.); connormbruce@ou.edu (C.M.B.)

2 European Centre for Medium Range Weather Forecasts, Reading RG2 9AX, UK; Peter.Bechtold@ecmwf.int (P.B.); Mark.rodwell@ecmwf.int (M.J.R.)

\* Correspondence: dparsons@ou.edu; Tel.: +1-405-325-8565

**Abstract:** Despite significant, steady improvements in the skill of medium-range weather prediction systems over the past several decades, the accuracy of these forecasts are occasionally very poor. These forecast failures are referred to as "busts" or "dropouts". The lack of a clear explanation for bust events limits the development and implementation of strategies designed to reduce their occurrence. This study seeks to explore a flow regime where forecast busts occur over Europe in association with mesoscale convective systems over North America east of the Rocky Mountains. Our investigation focuses on error growth in the European Centre for Medium-Range Weather Forecasting's (ECMWF's) global model during the summer 2015 PECAN (Plains Elevated Convection at Night) experiment. Observations suggest that a close, but varied interrelationship can occur between long-lived, propagating, mesoscale convection systems over the Great Plains and Rossby wave packets. Aloft, the initial error occurs in the ridge of the wave and then propagates downstream as an amplifying Rossby wave packet producing poor forecasts in middle latitudes and, in some cases, the Arctic. Our results suggest the importance of improving the representation of organized deep convection in numerical models, particularly for long-lived mesoscale convective systems that produce severe weather and propagate near the jet stream.

**Keywords:** Rossby wave trains; meoscale convective systems; forecast busts; error growth; medium-range numerical weather prediction; convective parameterization; PECAN field campaign

## 1. Introduction

It is now widely recognized that the skill of leading medium-range numerical weather prediction (NWP) systems has substantially advanced over the past several decades at a rate of approximately one day per decade [1]. This steady rate of advancement has been found to be due to improvements in the initial conditions and the formulation of the forecast model through scientific and technical efforts in the areas of data assimilation, model dynamics and physics, model resolution made possible by growing computational abilities, the characterization of uncertainties, and observing systems [2]. The advancement in the skill of these NWP systems have resulted in improvements to emergency preparation and disaster mitigation, including lives saved and a reduction in property damage, and the creation of substantial financial revenue across a variety of sectors of the economy [1]. The importance of accurate forecasts by these global models to society is illustrated by several examples including the World Economic Forum Surveys that lists extreme weather events and the related topics of natural

disasters, failure of climate change mitigation and adaptation, and water crises as the most likely risks to occur in the next ten years with the largest detrimental impacts on society [3].

Despite these steady advances in the skill of medium-range forecasts and the importance of these forecasts to society, there are occasions when these modeling systems produce forecasts with unexpectedly very poor skill. These events are referred to as "busts" or "drop-outs". Rodwell et al. (2013) [4] undertook a seminal study of busts in 22 years of simulations based on the fixed model utilized to produce the European Centre for Medium-Range Weather Forecasting (ECMWF) Reanalysis (ERA)-Interim.

Their study defined a bust when the 6 day forecast of 500 hPa geopotential heights over Europe had a root mean square error (RSME) greater than 60 m and an anomaly correlation coefficient (ACC) of less than 40%. This criteria ensures a relatively large error in both the magnitude of the 500 hPa heights and a significant discrepancy in the phase of any predicted feature in the flow. The Rodwell et al. study found that the composite initial conditions for busts over Europe were associated with a coherent flow pattern over North America with a trough over the Rockies with high convective available potential energy (CAPE) to the east. In their study, the composite conditions at the verifying time were associated with blocked flow (i.e., a high over northern Europe and a low over the Mediterranean) and an anticyclonic Rossby wave break over the Atlantic. The association between poor forecasts and anti-cyclonic wave breaking and the onset of blocking is expected as recent studies have shown that prediction of the onset of a blocking regime remains a challenge in global numerical models [5–7].

A subsequent climatological investigation of busts [8] again used the ERA-Interim as a fixed forecast model to go beyond composite conditions to explore different modes of forecast failure. This investigation found a seasonal variation in the dynamics of bust events and proposed that the events were related to the interaction of the Rossby wave guide formed by the mid-latitude jet stream with events characterized by strong diabatic heating. These events were proposed to include mesoscale convective systems (MCSs) over North America in the summer, recurring tropical cyclones in the fall, and upstream cyclogenesis during the winter. These two studies show that busts in the 6 day forecasts were associated with a change in the flow regime across the mid-latitudes and the Arctic (e.g., see Figures 10 and 12 in [8] and Figure 3 in [4]). These flow regime changes were consistent with a reversal in the sign of the North Atlantic oscillation (NAO) and occurred in the presence of a negative Pacific–North American (PNA) pattern index. The sensitivity of changes in the sign of the NAO to a negative PNA was previously known [9]. The linkage between Rossby wave breaking over the Atlantic and changes in the NAO is also long established [10]. Noting that Rodwell et al. found that forecast busts are associated with Rossby wave breaking, the association between wave breaking, large-scale regime change, and forecast busts, draws attention to the critical importance of reducing the frequency of medium-range busts to sub-seasonal to seasonal forecasts. Other factors supporting the need for research into this area are that the prediction of these blocking events at long lead times can have severe societal impacts [11] and the ensemble forecasts during busts can be overconfident [12].

Reducing the frequency of busts would be aided by an understanding of the dynamical evolution of the flow in these forecasts and what factors in the modeling system (e.g., observational strategies, data assimilation, model physics, resolution, dynamical core) need to be improved in order to reduce the frequency of these poor forecasts. In this regard, Grazzini and Isaksen (2002) [13] linked busts in the European Centre for Medium-Range Weather Forecasting (ECMWF) forecast model to MCS activity over the United States, particularly around the Great Lakes region. These forecast errors could be due to the interaction of MCS with the Rossby wave dynamics raising the possibility that the MCSs may slow the phase propagation of Rossby waves [4] or that diabatic heating near the jet stream could trigger Rossby wave packets that propagate downstream and subsequently amplify over the Atlantic Ocean [8]. The difficulty in obtaining skillful forecasts over Europe when Rossby wave packets are triggered over North American and break over the North Atlantic is consistent with the findings of low predictive skill with these short-track Rossby wave trains in contrast to higher predictive skill with longer-lived wave packets that can even traverse across the North Hemisphere

several times [14,15]. The impact of diabatic processes on disturbances within the wave guide and their influence on downstream high-impact weather was explored through the international North Atlantic Waveguide and Downstream Impact Experiment (NAWDEX) field campaign [16].

This study seeks to understand the processes that led to relatively poor medium-range predictions over Europe during June 2015, which coincided with the Plains Elevated Convection at Night Experiment (PECAN) field and modeling campaign [17]. PECAN was designed to advance our understanding of the mechanisms that maintain convection through the night over the Great Plains in an effort to explain the nocturnal maximum in deep convection over this region during the summer. Our efforts will focus on the initial generation and evolution of forecast errors that occur through the interaction of an MCS with a strong jet stream and associated wave guide properties. In selecting our approach, we note that Lorenz (1969) [18] cautioned against drawing conclusions from the superimposing errors and mean flow characteristics that are uncorrelated in predictability studies since in the atmosphere "*systems such as large cumulus clouds are not randomly distributed throughout the atmosphere, but have a preference for regions containing such meso-scale systems as squall lines and fronts. These in turn are not randomly distributed, but prefer certain locations relative to larger-scale synoptic features.*"

The scale interaction and error growth within the context of predicting large-scale atmospheric flow has been a topic of interest since the earliest days of numerical weather prediction [19]. The results of these studies include findings that errors at smaller scales processes can grow up-scale and reduce the predictive skill for larger-scale circulations [18,20], errors can spontaneously increase in magnitude across all scales [21,22], and the importance of large-scale errors interacting with smaller-scale features [19,23]. Numerous studies have also implicated convective processes as one of the initial sources of error in predictability studies, whether convection is partly parameterized or permitted to be resolved [24–29].

Zhang et al. (2007) [26] proposed a multi-stage conceptual model for how the errors in the treatment of convection influence the larger scales. In the first six hours of the simulation, errors in deep convection and the gravity waves generated by convection create strong, local errors. In the second stage that occurs 3 to 18 h into the simulation, these large errors associated with convection induce differences in potential vorticity that can induce errors in the balanced flow. Beyond approximately 12 h into the simulation, these errors begin to grow with the large-scale baroclinic instability. The error growth in this third stage is strongly dependent on background baroclinic waves. While this investigation by Zhang et al. (2007) filtered out wavelengths smaller than 1000 km, Tribbia and Baumherner (2004) [30] found that in a global climate model, the errors associated with the quasi-exponential growth of baroclinic disturbances had a spectral peak in the synoptic scales at relatively short wavelengths with between wavenumbers 10 and 20 (where the unitless wavenumber refers to the number of waves of a given wavelength required to encircle the earth at the latitude of the disturbance). Our investigation examines error growth from the interaction between MCSs and synoptic-scale waves and the subsequent error growth within the context of synoptic scale waves.

## 2. Data Set and Methods

The period of focus for this study corresponds to the Plains Elevated Convection At Night (PECAN) field campaign. This project took place from 1 June to 15 July 2015 with the goal of advancing knowledge of the mechanisms responsible for the initiation, generation, and maintenance of summer-time nocturnal MCS over the Great Plains (e.g., [17,31]). This large, multi-agency field campaign employed three research aircraft, a fixed S-band radar, an array of nine mobile scanning radars, fixed and mobile lower-tropospheric profiling systems, and numerous mobile, surface weather stations. The PECAN observing period was convectively active with the peak number of nocturnal MCSs per weak at 4, which is above average ([17]). The location of the highest number of nocturnal MCSs during PECAN, near 40–41 degrees latitude, is consistent with the climatological maximum in nocturnal MCSs activity (see Figure 1 in [17]). The location of the highest frequency of nocturnal MCS activity during PECAN and in the climatology is "upstream" and somewhat to the south of the

location where Grazzini and Isaksen (2002) [13] noted that MCS activity in the initial conditions was correlated with the highest frequency of busts.

Rodwell et al. [4] examined busts in 00 UTC forecasts of both the ECMWF operational high-resolution Integrated Forecast System (IFS), which is frequently upgraded to improve with time, and the fixed version of the IFS employed in creating the ERA-Interim data set. Their investigation noted that the number of busts in the operational modeling system decreased with time from 60 to 70 per year in the early 90s to approximately 5 in 2011. In contrast, the busts in the ERA-Interim systems decreased more slowly and generally remained between 20 and 40 busts per year. Employing the ERA-Interim as a fixed forecast model as in [4], but initialized at both 00 and 12 UTC, seven busts took place during the PECAN period (6 June at 00 UTC, 15 June at 00 UTC, 17 June at 12 UTC, 19 June at 00 UTC, 19 June at 12 UTC, 20 June at 00 UTC, 24 June 00 UTC). A single poor forecast near the bust criteria was also observed on the 26th of June. Given the annual number of forecast busts with the fixed forecast model noted in the Rodwell et al. [4] climatology, the period from 15 to 20 June seems to have low predictability and be particularly challenging for the version of the IFS utilized in the ERA-Interim reanalysis.

The root mean square error and the anomaly correlation coefficient of the 500 hPa heights were calculated utilizing the operational version of the IFS during this June 2015 period as shown in (Figure 1). Comparing the bust dates for the reanalysis and the operational systems reveal the general improvement in the operational system consistent with the findings of Rodwell et al. (2013). For example, while 7 busts took place in the ERA-reanalysis forecasts, only four forecasts utilizing the ECMWF operational system in 2015 had ACC values below the 0.4 cutoff with a 5th value near the cutoff. While numerous forecasts had RMSE values above the 60 m cutoff for a bust, the correspondence between the two criteria meant that only the operational forecast on 20 June would have been classified as a bust with both the ACC and RMSE criteria being met. A second case, 12 UTC on 22 June was also quite close to a bust event with a RMSE value quite close to the bust criteria. The general pattern of the ACC scores (Figure 1) suggests lower skill at 12 UTC on the 13, 14, and 15th of June and from the 20th to the 23rd of June. The lack of a close correspondence between the dates of the busts in the operational ECMWF IFS and ERA-Interim forecasts is expected given that the two systems utilize different assimilation systems, resolution, and model physics. The lack of a one-to-one correspondence in the bust dates also hints at challenging complexities faced in efforts to reduce busts and advance predictive skill.

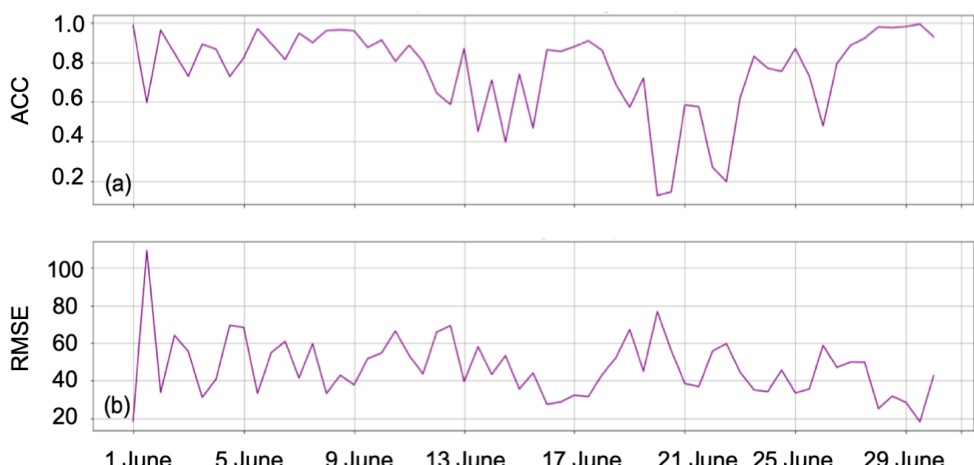

**Figure 1.** Measures of skill for 6 day forecasts for 500 hPa geopotential height over Europe made by the European Centre for Medium-Range Weather Forecasting (ECMWF) Integrated Forecast System. The abscissa denotes the time when the forecast was initialized. (**a**) Anomaly correlation coefficient (ACC), (**b**) the root mean square error (RMSE) (m).

To diagnose the source, propagation, and growth of the error in the forecast, we follow the method outlined in [32] so that a phase-independent error amplitude is derived from a linear combination of terms proportional to wave energy and wave enstrophy in the horizontal plane, using a streamfunction calculated from the horizontal wind forecast error field:

$$E = \frac{1}{2}\left[\left(\frac{\partial \psi_e}{\partial x}\right)^2 + \left(\frac{\partial \psi_e}{\partial y}\right)^2 - \psi_e\left(\frac{\partial^2 \psi_e}{\partial x^2} + \frac{\partial^2 \psi_e}{\partial y^2}\right)\right], \tag{1}$$

where $E$ is the total error amplitude, with units of $m^2/s^2$, and $\psi_e$ is the error streamfunction calculated from $u_e$ and $v_e$. Considering a packet of error in the form of a plane wave given by $\psi_e = \psi_{e0}sin(kx + ly + \phi)$, the error amplitude simplifies to $E = \frac{1}{2}\psi_{e0}^2(k^2 + l^2)$. This metric thus refers to the amplitude of rotational errors, and specifically appeals to features at the larger end of the mesoscale regime into synoptic and planetary scales. To calculate this phase-independent error amplitude, operational forecasts from the ECMWF IFS were accessed from the THORPEX Interactive Grand Global Ensemble (TIGGE) [33] at a 1 degree by 1 degree spatial resolution and a 6 hour time resolution. The error field is then calculated based on the difference between the ECMWF IFS forecast and the corresponding operational analysis also obtained from TIGGE.

Given the conclusions of [4] and [8] that forecast busts over Europe are associated with Rossby wave dynamics and that the structure and evolution of synoptic-scale error resemble Rossby Wave Packets [34], it is insightful to formulate the errors in a wave and wave packet framework. Based on the derivation of a phase-independent wave-activity flux by [35] for stationary [35] and migratory [36] eddies on a zonally-varying base state, an error wave-activity flux is formulated for an error streamfunction perturbation given by $\psi_e$,

$$\vec{W_e} = \frac{1}{2|\vec{U}|}\begin{pmatrix} U\left[\left(\frac{\partial \psi_e}{\partial x}\right)^2 - \psi_e\frac{\partial^2 \psi_e}{\partial x^2}\right] + V\left[\frac{\partial \psi_e}{\partial x}\frac{\partial \psi_e}{\partial y} - \psi_e\frac{\partial^2 \psi_e}{\partial x \partial y}\right] \\ U\left[\frac{\partial \psi_e}{\partial x}\frac{\partial \psi_e}{\partial y} - \psi_e\frac{\partial^2 \psi_e}{\partial x \partial y}\right] + V\left[\left(\frac{\partial \psi_e}{dy}\right)^2 - \psi_e\frac{\partial^2 \psi_e}{\partial y^2}\right] \end{pmatrix} + \vec{C_{eu}}M_e, \tag{2}$$

$$M_e = \frac{E}{2|\vec{U} - \vec{C_{eU}}|}, \tag{3}$$

where $\vec{U} = \frac{1}{2}(\vec{U_f} + \vec{U_t})$ is the average of the forecast wind $\vec{U_f}$ and the true wind $\vec{U_t}$, and $U$ and $V$ are their zonal and meridional components respectively. $M_e$ is the pseudomomentum of the error field, adapted from [36], and proportional to the error amplitude (Equation (1)) scaled by the magnitude of the base state Doppler-shifted wind (Equation (3)). For migratory waves, $\vec{C_{eU}}$ is the phase velocity vector of the error wave in the direction of the mean wind. The base state upon which errors propagate is thus given by the mean of the forecast and the truth. Note that these two fields diverge as lead-time increases, and their average will move toward a filtered low wavenumber pattern.

For the same plane wave error packet as above, the error wave activity flux can be written as

$$\vec{W_e} = \frac{\psi_{e0}^2}{2|\vec{U}|}\begin{pmatrix} Uk^2 + Vkl \\ Ukl + Vl^2 \end{pmatrix} + \frac{\psi_{e0}^2|K|^2}{4|\vec{U}|}\left(\frac{|\vec{U}||K|^2}{|\nabla Q|} - 1\right)\vec{U}, \tag{4}$$

$$\vec{W_e} = \vec{C_{eg}}M_e = \vec{C_{eg}}\frac{E}{2|\vec{U} - \vec{C_{eU}}|}, \tag{5}$$

for stationary Rossby waves in a zonally-propagating Rossby wave packet, this simplifies to $\vec{W_e} = E\frac{\vec{C_{eg}}}{|\vec{C_{eg}}|}$, i.e., the error wave activity flux is a vector in the direction of the error group velocity with the magnitude

of the error amplitude. Packets of error are characterized by local maxima in *E*, with $\vec{W_e}$ oriented parallel to their group velocity.

## 3. Results

### 3.1. Overview of Synoptic Flow

A Hovmoeller diagram of both the zonal and meridional winds at 250 hPa over the middle latitudes for the June 2015 is shown in Figure 2. Several features are evident including a period of strong zonal winds from 15 to 23 June over a longitude (260 to 300) that roughly correspond to the longitude of central North America to the western Atlantic and the northeastern portions of the continent (Figure 2). This period of strong zonal winds includes several jet streaks with enhanced zonal winds and also includes the periods with decreased forecast skill discussed earlier. The meridional winds at 250 hPa (Figure 2b) shows that the flow is generally characterized by the well known eastward propagation of synoptic-scale Rossby wave packets with both the phase speed of the individual peaks in the meridional winds and the downstream propagation of the wave packet with the group velocity both evident. However, the pattern of meridional winds undergoes a distinct change beginning near the time of strong zonal flow and lowered forecast skill with streaks of northerly and southerly flow extending across the Atlantic indicated a transition to waves with a rapid phase speed (Figure 2b). Near and after this time there is a tendency for the waves to move through a more stationary flow pattern. For example, persistent northerly and southerly flow develops near 20 and 40 degrees longitude, respectively. A tendency for a more stationary pattern in the meridional flow also becomes evident over the Pacific Basin (e.g., southerlies just west of 180 degrees longitude) and after 25 June over the central to western North America (160 to 260 degrees longitude). Superimposed on this more stationary pattern are active smaller-scale synoptic disturbances.

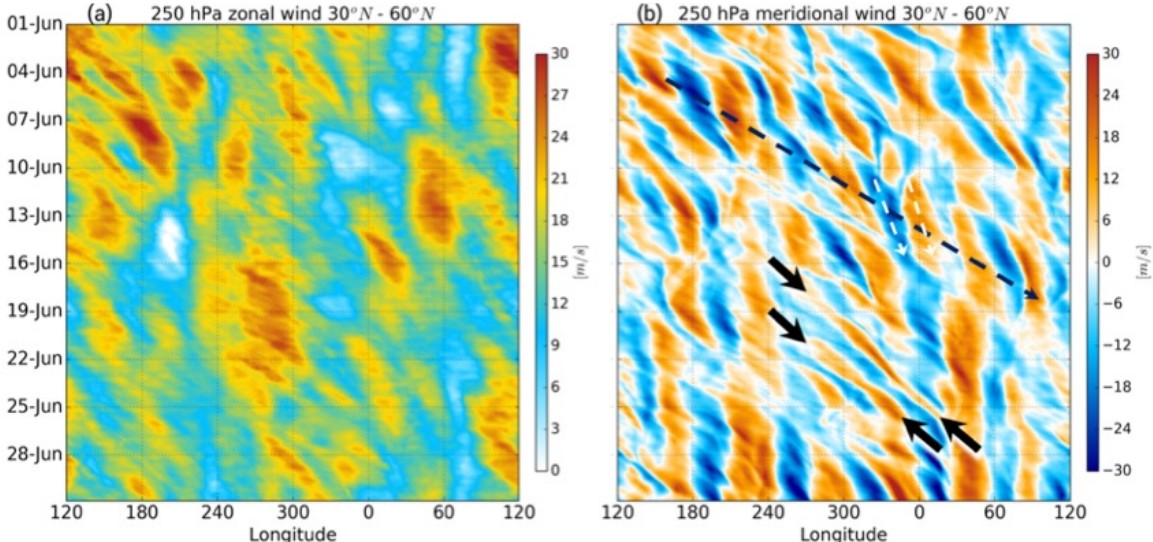

**Figure 2.** Hovmoller plot of 250 hPa winds averaged between 30° N and 60° N from ECMWF Reanalysis (ERA)-5 reanalysis [37] for June 2015. (**a**) Zonal wind component, (**b**) meridional wind component with the dashed line representing a qualitative designation of a Rossby wave packet moving downstream with the group velocity, the dotted white line corresponding to the phase speed of the individual troughs and ridges, and the black arrows indicating rapidly propagating individual disturbances.

Spectral analyses of the meridional winds at 250 hPa confirms the change in the behavior of the Rossby wave packet (Figure 3). In this spectral analysis, the flow over the 10 day period from 6 to 15 June is characterized by wavenumbers between 5 and 10 with a peak near wave number 8 (Figure 3a). These wavenumbers are within the range commonly found in middle latitude flows, as for example [38] found

that Rossby wavenumbers between 6 and 9 dominate during middle latitude summers. The wave energy shifts dramatically to higher wave numbers of between 8 and 13 (Figure 3b) for 18 to 27 June. This time range included the prolonged period of lower ACCs in Figure 1. As mentioned earlier, Tribbia and Baumherner (2004) [30] linked higher wavenumber baroclinic disturbances to rapid error growth. These higher wavenumber disturbances are (Figure 3b) also associated with rapid movement of the wave packet as evidenced with a change in the frequency from centered on approximately 0.1 for the period from 6 to 15 June to a frequency of just below 0.3. This higher frequency implies that the period for disturbances would approach 2–4 days rather than the 7 to 10 days with the more typical synoptic-scale waves. The power spectra for June over the years 1979–2018 (Figure 3c) reveals that slow moving wavenumbers (4 to 9) are the most commonly occurring wave disturbance consistent with [38], while Figure 3d suggests that the rapidly moving waves observed during 18 to 27 June were a relatively rare occurrence.

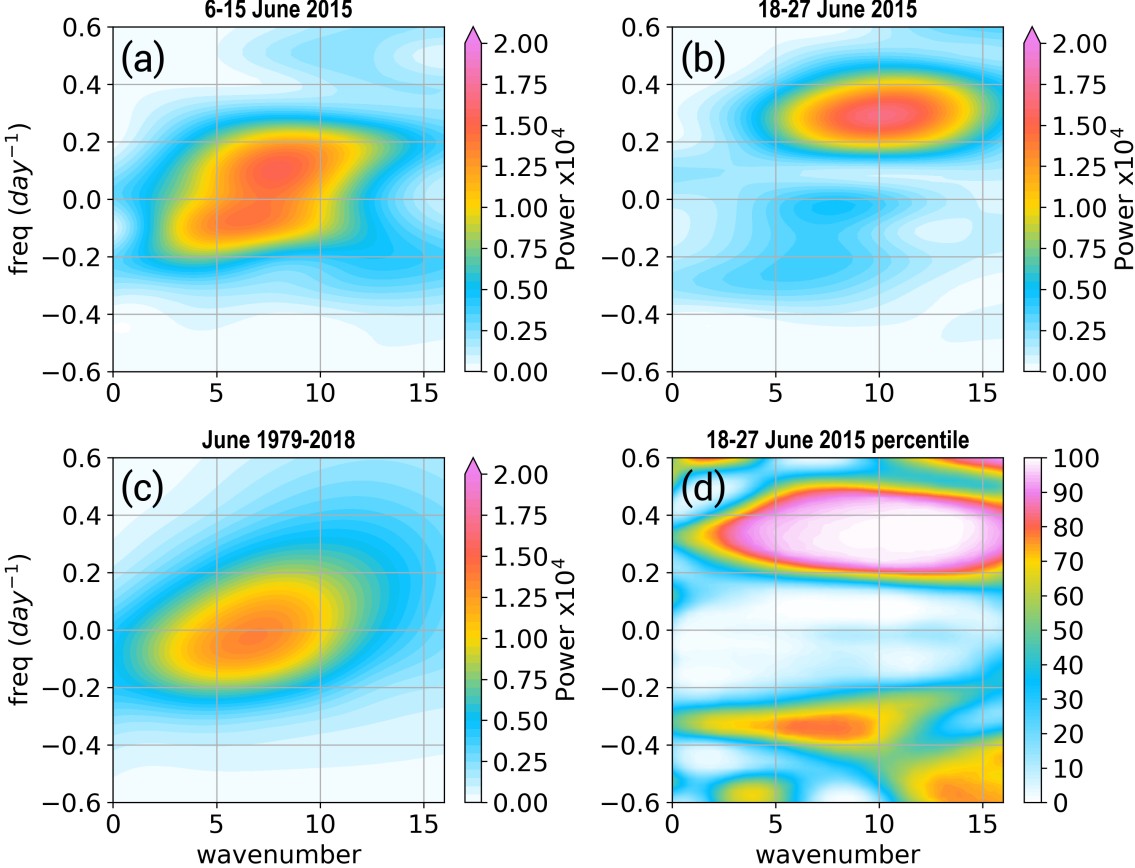

**Figure 3.** Spectral analysis of the meridional winds at 250 hPa from the ERA-5 reanalysis [37] averaged over 30 to 60 N latitude and 270 to 360 E longitude. (**a**) For the period of 6 to 15 June 2015, (**b**) for the period of 18 to 27 June 2015, (**c**) for climatology as indicated by June 1979–2018, (**d**) the percentile of the June climatology in the power spectra for the 18 to 27 June 2015 period.

Insight into the flow regime is also obtained from the average 250 hPa winds (Figure 4a) that reveal a strong jet over near the border between Canada and the U.S. associated with a merger of the middle latitude jet stream and strong flow diving southward from the Arctic north of Alaska. This strong jet over North America associated with this merger is consistent with the pattern of stronger westerly winds in Hovmoeller diagram shown earlier (Figure 2). Upstream of the flow over North America, a strong jet is also present over the central Pacific, while downstream over the eastern and central Atlantic, the jet stream is weaker. The corresponding flow at 850 hPa (Figure 4b) reveals that the strong coherent flow pattern originating and returning to the Arctic that was present at 250 hPa also exists at

lower levels. Over the Great Plains of the U.S., there is southerly flow at 850 hPa so that the area near the Canadian border where these two flow regimes intersect has significant baroclinicity (not shown). Over the central Pacific and central Atlantic, the 850 hPa flow has similar characteristics as the jet aloft, but with the SW-NE oriented appearance of a warm conveyor belt.

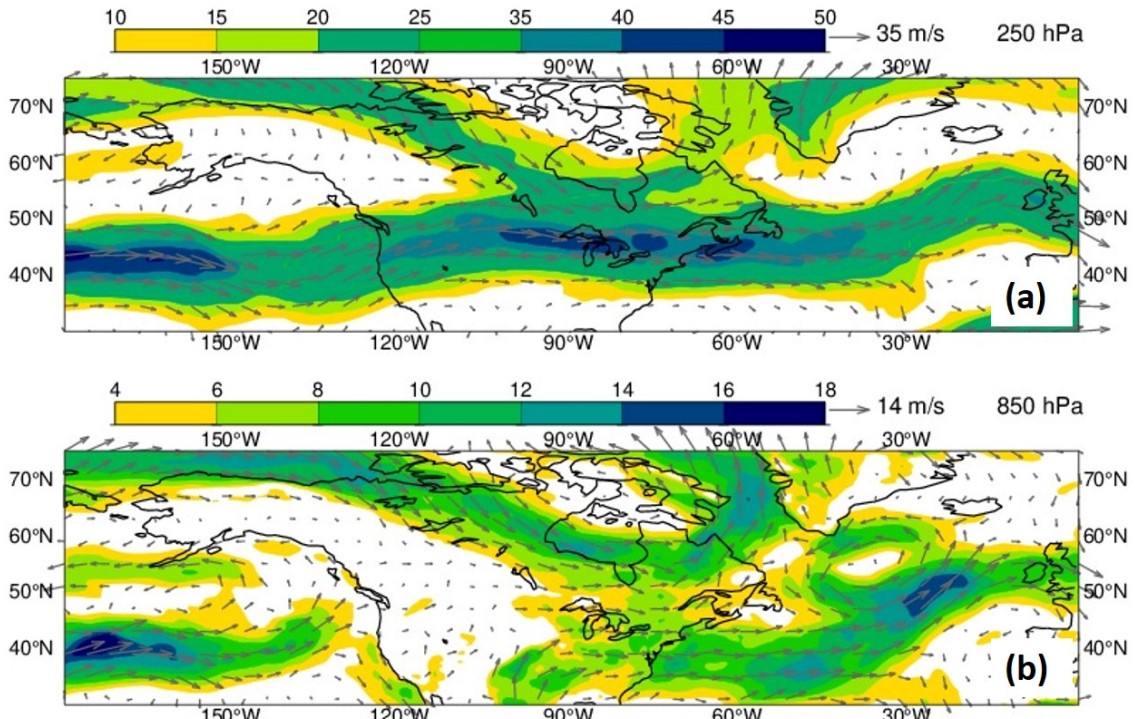

**Figure 4.** Winds from the ERA-5 reanalysis [37] averaged over the time period from 18 to 24 June 2015. (**a**) Winds speed (m s$^{-1}$) and contoured at the 250 hPa level to show the regions of peak flow. (**b**) As in (**a**) but for the 850 hPa level.

## 3.2. Interaction between Convective Systems and Rossby Wave Packets

Rodwell et al. (2013) [4] proposed that one regime where forecast busts occur is when a trough is over the Rockies and convective systems form to the east. Their study suggests that a reduction in the frequency and magnitudes of busts might occur through the reduction of errors from more accurate initial conditions, advances in the model physics, and improvements in the representation of forecast uncertainty. On the basis of a potential vorticity budget, Rodwell et al. noted that the role of the MCS and diabatic processes was important and in general slowed the propagation of the trough. We will begin with an examination of the behavior of the convection and the Rossby wave structure for periods with the low and high number wavenumber disturbances shown earlier in Figure 3.

### 3.2.1. 11–12 June 2015

Our investigation of the period that was characterized by the more typical lower wavenumber Rossby waves (Figures 2 and 3b) is focused on the convection that took place on 11 and 12 June 2015. The forecasts initialized on these days were not busts, but were associated with the beginning of a period of declining skill in the ECMWF IFS, as suggested from the changes in the ACC (Figure 1). The upper-level flow for this case, as evident by the wind speeds at 250 hPa (Figure 5), shows ridging over the central Plains with a highly disturbed polar jet stream with multiple jet streaks. There is also evidence for a subtropical jet stream entering North America along the west coast with a weak trough over Baja California. The flow is relatively complex as the two jet streams evolve toward merging over the northern Plains during this period (Figure 5).

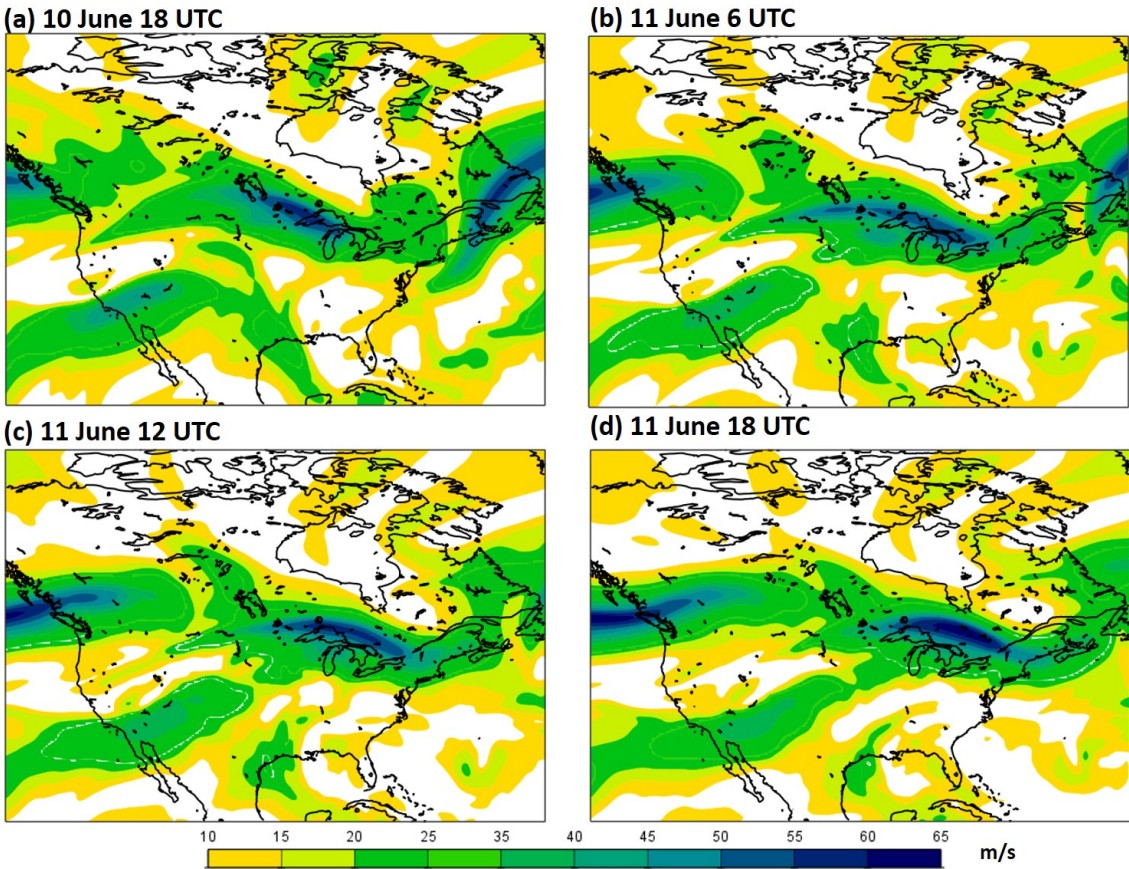

**Figure 5.** Wind speed (m s$^{-1}$) contoured at the 250 hPa level from the ERA-5 reanalysis [37] for 10–11 June 2015. (**a**) 18 UTC on 10. (**b**) 6 UTC on 11 June. (**c**) 12 UTC on 11 June. (**d**) 18 UTC on 11 June. The velocity scale is shown at the bottom of the plot.

The evolution of the meridional winds from the ERA-5 reanalysis [37] and reflectivity from the operational radar network are shown in Figure 6 for 00 UTC on 11 June to 04 UTC on 12 June 2015. Several inferences can be drawn from this analysis including that the strong southerly winds remain relatively close to the MCS and increase in strength from 15 to 25 m s$^{-1}$ as the MCS intensifies and grows up-scale (Figure 6). The proximity of the strong southerlies aloft at 250 hPa to the MCS suggests the possibility that the outflow from the MCS is interacting with the southerly flow ahead of the trough in the Rossby wave. The MCS on 11–12 June is the focus of a recent paper by Zhang et al. [39]. According to their observational analysis and accompanying simulation, this MCS was associated with lifting of a moist, southerly flowing air mass resulting in a strong front-to-rear inflow, which with the orientation of the MCS generate strong southeasterly flow near the top of the storm. Hence, the strong southerly winds in the vicinity of the MCS are likely associated with the outflow aloft. The observations and simulations in that study [39] also suggest that the MCS formed near a stationary front and propagated ahead of the front during the night with the MCS maintained by a bore that was initiated by the lifting of stable layer over a convectively-generated cold outflow. This lifting greatly reduced the convective inhibition in the inflow air mass allowing the MCS to be maintained during the night even as the boundary layer stabilized. The varying role of the cold front, convectively-generated cold outflows, and bores in initiating and maintaining the MCS in the Zhang et al. study [39] reinforces the challenge and complexity in parameterizing long-lived continental MCS in numerical weather prediction models.

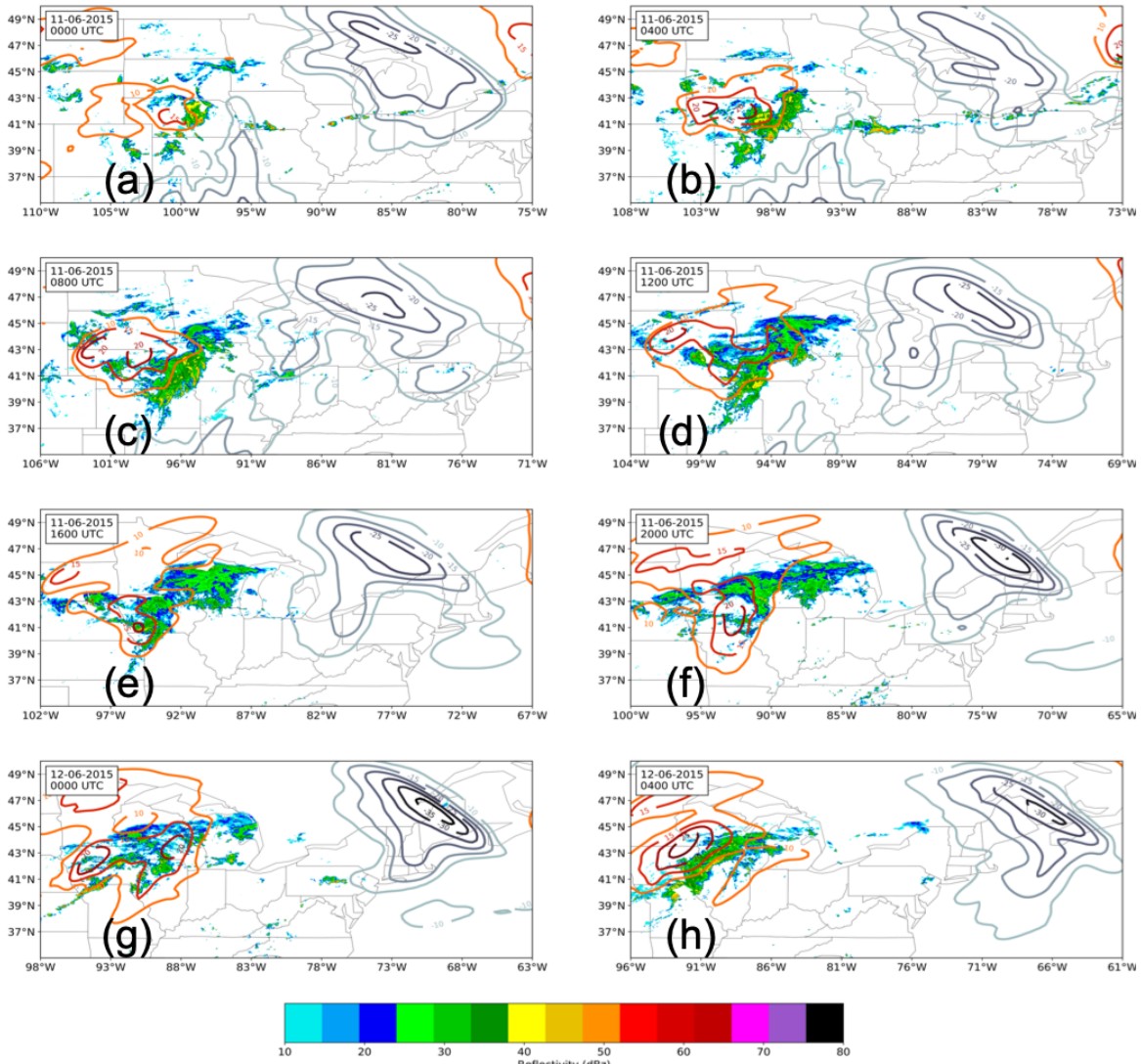

**Figure 6.** Meridional winds (m s$^{-1}$) at the 250 hPa level from the ERA-5 reanalysis [37] (color contours with red southerly and blue northerly winds, respectively) and the radar reflectivity in dBZ with the scale shown at the bottom of the figure. (**a**) 00 UTC on 11 June. (**b**) 04 UTC on 11 June. (**c**) 08 UTC on 11 June. (**d**) 12 UTC on 11 June. (**e**) 16 UTC on 11 June. (**f**) 20 UTC on 11 June. (**g**) 00 UTC on 12 June. (**h**) 04 UTC on 12 June 2015.

Another inference from Figure 6 is that the northerly winds at the 250 hPa level ahead of the southerlies also intensify with the maximum northerly winds increasing from 20 m s$^{-1}$ to over 35 m s$^{-1}$. As a results of these increases in the meridional winds, the ridge in the jet stream has amplified. These northerlies with the ridge move downstream more rapidly than both the the MCS and the southerly flow. The Rodwell et al. study [4] proposed that the interaction between convection and the pre-existing ridge would intensify the wave and "virtually halt the eastward progression of the trough". The behavior of the meridional flow (Figure 6) suggests the southerlies ahead of the trough are propagating eastward much slower than the northerlies consistent with the Rodwell et al. study, again suggesting a slowing in the phase speed of the trough through the impacts of diabatic processes. An examination of the northerly flow reveals an area of northerly winds forming at nearly similar latitude as the convective system located, but located closer to the MCS and southwest of the stronger, pre-existing northerly flow (i.e., see panels of 0800 and 1200 UTC on 11 June in Figure 6). This secondary area of northerlies was transient and weakened as the longer wavelength disturbance came to dominate the flow.

The interaction of the MCS and the jet stream is also evident by the changes in the zonal wind at 250 hPa (Figure 5). From this figure, there is a modest increase (10 m s$^{-1}$) in the velocity of the jet streak over the northern Plains from 18 UTC on 10 June through 18 UTC on 11 June as this zonal wind maximum propagated to the east. From a comparison of the location of the strengthening meridional flow (Figure 6) and the weakly intensifying jet (Figure 5), the intensification zonal flow falls between the intensification of the southerly and northerly flow and appears to be related to the impact of convection on the jet dynamics. This jet streak forming downwind of the convective outflow is expected and somewhat consistent with the second stage in error growth discussed by Zhang and Bei (2007) [40] as the convective outflows transitions to balanced flow.

The additional challenge of treating the interaction of the convective systems with the jet stream and Rossby wave dynamics can be seen in Figure 7. This figure shows the streamlines at 250 hPa, along with the error amplitude and error in the wave activity flux. In the 24 h forecast (Figure 7a), the largest error amplitudes in the western hemispheric view in this figure occurs in the ridge near the Great Lakes in the vicinity of the organized convection. At 36 h into the forecast, the error amplitude continues to increase in that area and remains the largest errors in the western hemisphere (Figure 7b). This finding suggests that the ECMWF IFS model may not be capturing the observed amplification of the ridge. In addition, the error in the wave activity flux appears in the 36 h forecast suggesting the downstream propagation of a Rossby wave response along the jet has developed at this time. At 72 h into the forecast (Figure 7c) the streamlines suggest an amplified wave pattern over North America and little movement in the trough near the border between the northeast US and Canada. The magnitude of the error amplitude and the spatial extent of the error in the wave activity flux at this time have increased. Small errors are also evident over portions of northern Europe and the northwest Atlantic. The error in the forecast at 84 h (Figure 7d) shows this downstream propagation and amplification of the errors continue over the North Atlantic with the largest error amplitudes in the observed trough and extending into the downstream ridge. The error in the wave activity flux (Figure 7) again shows the wave response is moving downstream through the ridge located over the central North Atlantic.

These results suggest that the MCS is associated with a slowing of the trough and an amplification of the Rossby wave pattern initially with the ridge. The initial error occurs within the ridge downstream of the convection. The subsequent movement of the error toward the east at a speed greater than that of the phase speed of the existing ridges and troughs in the flow is consistent with concept of the error moving with the group velocity of Rossby waves. Estimating this eastward movement of the error using the location of the error amplitude near the Great Lakes in the 24 forecast to over Ireland in the 84 h forecast in Figure 7 and using a mean latitude of 48 N leads to an estimate of the error propagation of approximately 32.4 m s$^{-1}$. This value is relatively close to the speed of the error propagation over the North Pacific of 31 m s$^{-1}$ found in [34] and close to the values given [41].

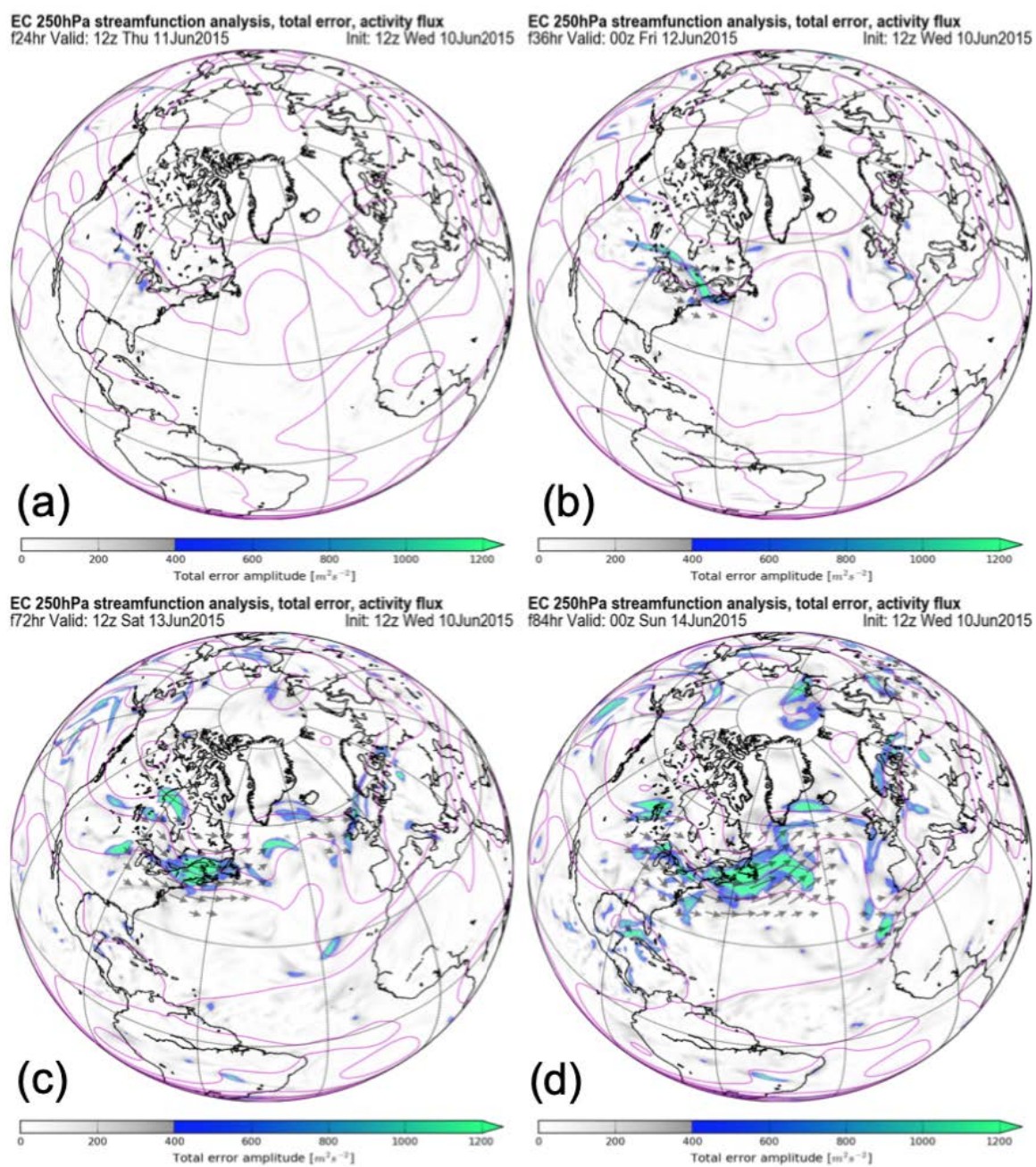

**Figure 7.** Streamlines, error amplitude, and error in the wave activity flux for the ECMWF IFS forecast initialized at 12 UTC on 10 June 2015 for (**a**) 24 h, (**b**) 36 h, (**c**) 72 h, and (**d**) 84 h forecasts.

### 3.2.2. 20 June 2015

The ACC and RMSE for 20 June 2015 forecasts with the operational ECMWF IFS had low skill with the forecast initialized on 000 UTC meeting the criteria for a bust (Figure 1). The background winds at 250 hPa for this case (Figure 8) has several differences from the upper-air flow on 11 June (Figure 5). For example on 20 June (Figure 8) there is a broad area of strong westerlies along the west coast, a tendency for a trough over the Rocky Mountains, and a merger of the middle latitude and flow originating over the Arctic. A trough over the Rocky Mountains has been implicated in busts in the medium-range forecast over Europe [4].

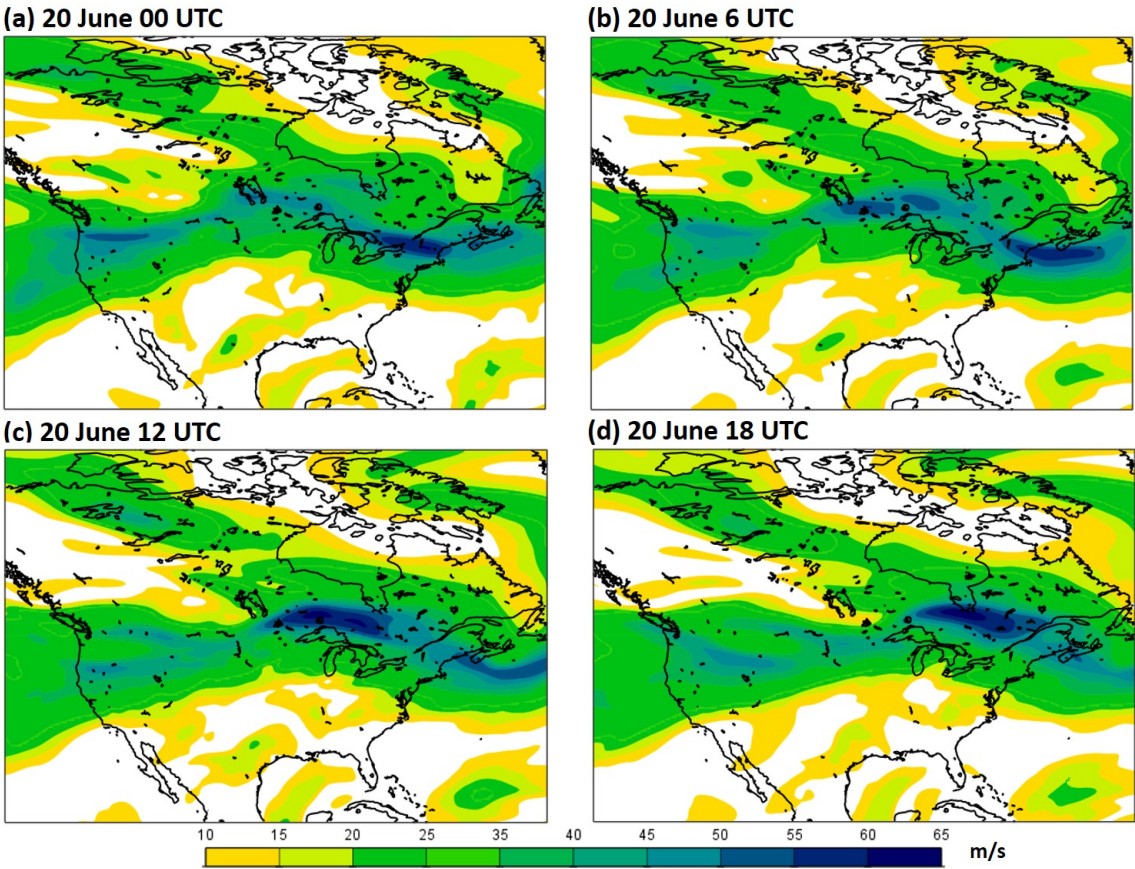

**Figure 8.** Wind speed (m s$^{-1}$) contoured at the 250 hPa level from the ERA-5 reanalysis [37] for 20 June 2015. (**a**) 00 UTC, (**b**) 06 UTC, (**c**) 12 UTC, and (**d**) 18 UTC on 20 June.

The radar reflectivity and meridional winds from the ERA-5 reanalysis [37] are shown in Figure 9. This case has several similarities to the analysis produced for the 11 June case (Figure 6). For example, the southerly winds remain located relatively close to the MCS over its lifetime with the peak southerlies generally located to the north or north-northwest the MCS (Figure 9). The northerly flow also appeared linked to the MCS with northerly flow generated on the jet stream downwind of the MCS (e.g., see evolution from 08 UTC to 16 UTC on 20 June in Figure 9). This generation of northerly flow occurred at a location upstream of the existing northerlies again raising the possibility that the MCS was forcing a shorter wave length disturbance. However, this "shorter-wavelength" ridge is more persistent over time for the 20 June case than the 11 June case as evidenced from a comparison of Figures 6 and 9.

The surface pressure and associated pattern of CAPE in the forecasts for 20 June (Figure 10) also suggests that the linkage of the MCS and synoptic-scale disturbances. In particular, the surface low is associated with the advection of highly unstable air toward the low also supporting any MCS that develops in association with this system. The magnitude of the CAPE is significant, especially considering that the forecast verification time is 06 UTC (midnight local standard time). The low pressure center formed along a stationary front. The 11 June MCS was also associated with the development of a surface low pressure center that formed along a stationary front (not shown). In contrast, Tropical Storm Bill evidenced by the low pressure system in the Mississippi valley is not associated with CAPE favorable for deep convection at this time (Figure 10). Earlier in its lifetime, this system produced significant flooding across Texas and Oklahoma. In general, Tropical Storm Bill was not associated with a signal in the meridional winds until on 21 June (Figure 9) when southerlies developed as the storm approached the jet stream. While the southerlies were weaker than the peak disturbance associated with MCS and the downstream northerlies had already begun to intensify,

Tropical Storm Bill reinforced the southerlies and may have also played a role in the intensification of the downstream northerlies later in the period.

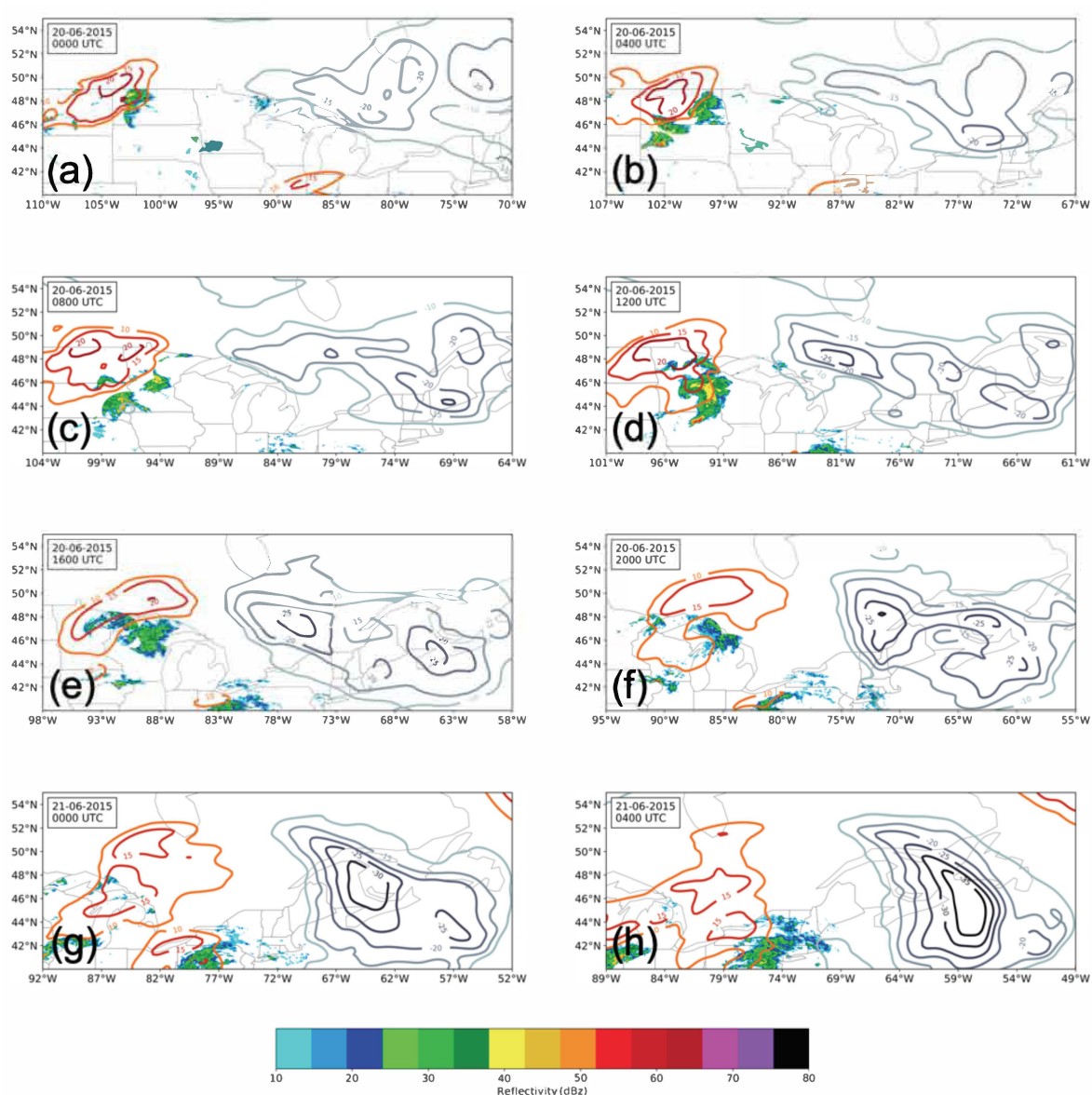

**Figure 9.** As in Figure 6, but with (**a**) 00 UTC on 20 June. (**b**) 04 UTC on 20 June. (**c**) 08 UTC on 20 June. (**d**) 12 UTC on 20 June. (**e**) 16 UTC on 20 June. (**f**) 20 UTC on 20 June. (**g**) 00 UTC on 21 June. (**h**) 04 UTC on 21 June 2015.

A difference between the MCSs on these 11 and 20 June cases is the relatively rapid movement of the first of multiple MCSs on 20 June with the system moving from the upper Great Plains on 00 UTC 20 June to the middle Atlantic States along the East Coast on 8 UTC on 21 June 2015 (Figure 9). The evolution of the zonal velocity at 250 hPa is also quite different between the 11 June (Figure 5) and 20 June convective events (Figure 8) with a marked increase in the velocity of the jet streak by order of 15 m s$^{-1}$ in just an 18 h period from 00 UTC to 18 UTC on 20 June. The peak in the zonal winds that formed between two southerly and northerly wind maximum evident at 00 UTC on 20 June (Figure 8) is consistent with a reduction of the wave length of the meridional flow on the jet stream that occurred with the development of northerlies just downstream of the MCS (Figure 9).

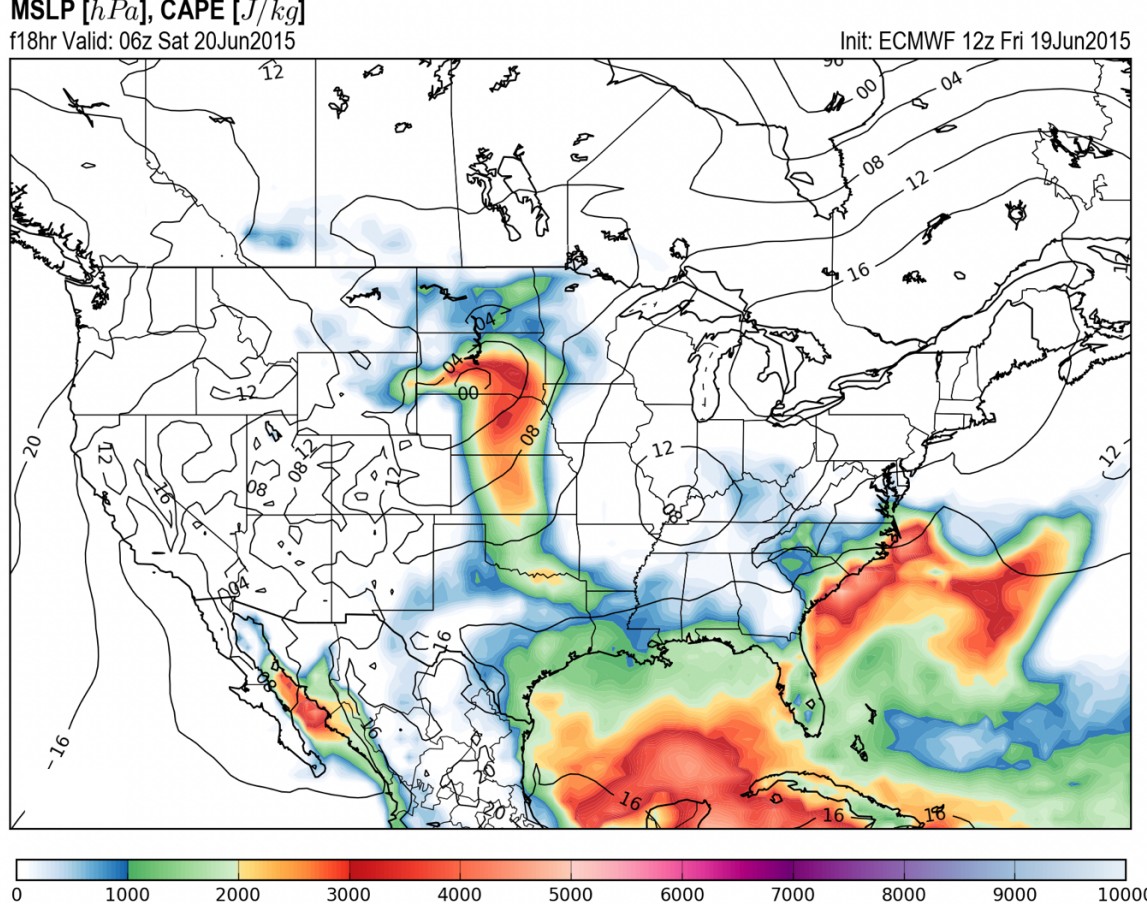

**Figure 10.** 18 h forecast by the ECMWF IFS initialized at 12 UTC on 19 June and valid at 06 UTC on 20 June. The Convective Available Potential Energy and sea-level pressure are shown.

At the same time as this intensification of the shorter wavelength northerlies and the jet streaks develop downstream of the MCS, the peak zonal winds associated with the pre-existing upper-level flow decreases in magnitude as evidenced by the decrease in the magnitude of the jet streaks over the Pacific northwest and New England with time in Figure 8. The meridional wind at the 250 hPa from the ERA-5 reanalysis [37] shown for the period from 00 UTC on 20 June to 12 UTC on 23 June 2015 (Figure 11) also supports the evolution toward higher wavenumber Rossby waves. The time period of Figure 11 corresponds to when the short wavelength (high wavenumber) pattern is becoming well established (Figure 3), but initially less well defined. This complexity in the Rossby wave pattern was noted earlier with the general change in the meridional winds shown in the Hovmoeller diagram for June (Figure 2). The phase speed of the shorter wavelength disturbance can be estimated from Figure 11. For example, tracking the southerly flow from approximately 105 W at 00 UTC on 20 June to 67.5 W on 00 UTC 21 June at 00 UTC on 21 June leads to a phase speed of 30.7 m s$^{-1}$. This speed is close to the speed of the error propagation found on 11 June. This rapid phase speed is consistent with the phase speed in Hovmoeller diagram shown earlier (Figure 2) and surprisingly close to the group velocity of more commonly occurring synoptic-scale Rossby waves in Figure 3.

The streamlines at 250 hPa, the error amplitude, and the error in the wave activity flux are shown in Figure 12 for the ECMWF IFS forecast initialized on 00 UTC on 20 June. Once again in the 24 h forecast, the largest error amplitudes evident in the western hemispheric occur in the ridge in the vicinity of the organized convection. The error at 36 h (Figure 12b) is again evident as an error in the wave activity flux with a downstream propagation along the jet. Consistent with the results shown in Figures 2 and 3, the downstream propagation of the error is far more rapid in this short wave regime with the largest error amplitude and error in the wave activity flux associated with the

downstream trough over the Atlantic. At 72 h, additional MCS activity over North America, which will be discussed in the next section, is associated with an increase in the errors over North America (Figure 12). Downstream at this time, the error amplitude, associated error in the wave activity flux, and trough continue to grow and rapidly propagate downstream with the error in the wave activity flux nearly reaching Europe in just 72 h (Figure 12c). At 96 h (Figure 12d), the error patterns have merged and expand in magnitude and extent so that large portions of the North Atlantic have errors in the wave activity and large values of the error amplitude. The northeastern orientation of the error in the wave activity flux over the North Atlantic with polar directed energy flux also suggests the forecast has trouble representing an anticyclonic wave break. The errors in the wave activity flux at this time also expanded in terms of spatial coverage with the leading edge having reached into eastern Europe. From this pattern, it appears that the error in the wave activity flux is moving at a more rapid pace than the troughs and ridges in the flow that indicate the phase velocity of the wave.

The 20–21 June case has some similarities to the previously discussed 11–12 June event. In both cases, the meridional winds seem to intensify with the MCS with the southerlies located close to the MCS during most of its lifetime and northerlies intensifying upstream. These shorter wavelength northerlies were more sustained in the 20–21 June event. The ECMWF had errors in the vicinity of the ridge suggesting possible errors in representing convection and its impact on the flow. The errors in both cases moved downstream at a rate faster than the propagation of the phase speed of the individual troughs and ridges. The nature of the movement of the MCS, wave pattern, and error propagation were quite different between the two cases with higher speeds in the 20–21 June case.

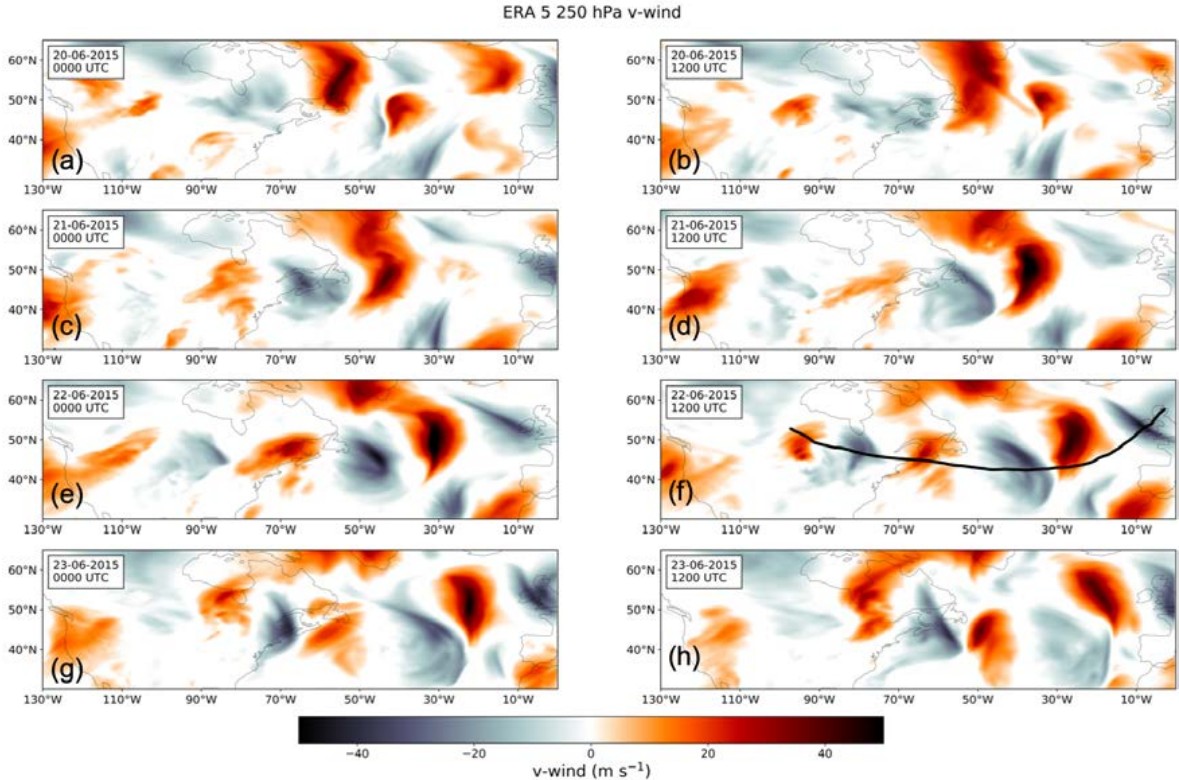

**Figure 11.** Depiction of the meridional wind (m s$^{-1}$) from the ERA-5 reanalysis [37]. (**a**) 00 UTC on 20 June. (**b**) 12 UTC on 20 June. (**c**) 00 UTC on 21 June. (**d**) 12 UTC on 21 June. (**e**) 00 UTC on 22 June. (**f**) 12 UTC on 22 June. (**g**) 00 UTC on 23 June. (**h**) 12 UTC on 23 June 2015. during the period of 00 UTC on 20 June through 12 UTC on 23 June 2015. The short wavelength Rossby wave packet is designated with a fine black line in the panel for 12 UTC on 22 June.

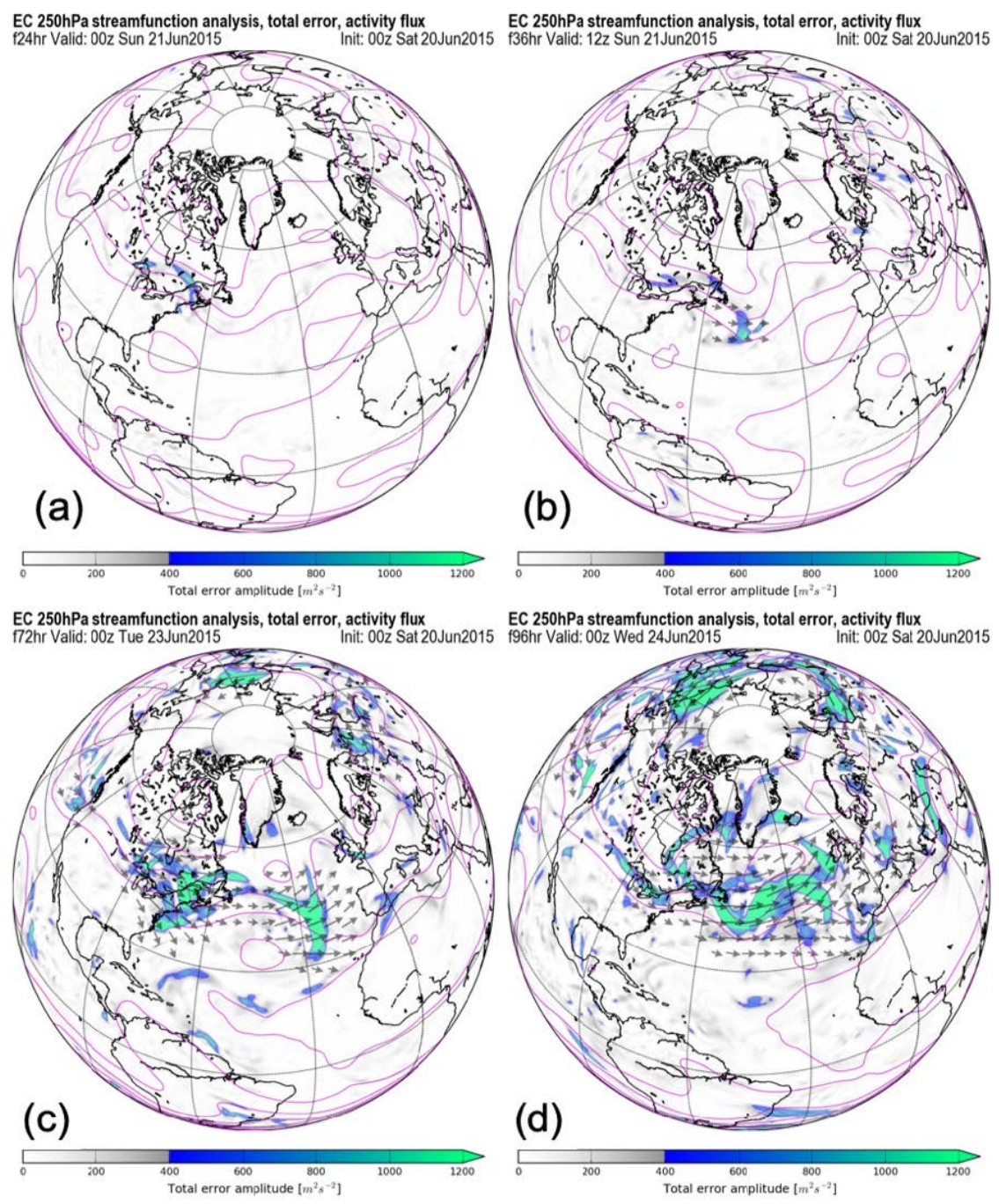

**Figure 12.** Streamlines, error amplitude, and error in the wave activity flux for the ECMWF IFS forecast initialized at 00 UTC on 20 June 2015. (**a**) 24 h forecast, (**b**) 36 h forecast, (**c**) 72 h forecast, and (**d**) 96 h forecast.

## 4. Discussion

### 4.1. Initial Forecast Errors

This study confirms several aspects of previous work. For example, the initial errors were in the vicinity of organized convection and it has long been known that rapid error growth occurs on the mesoscale with moist processes (e.g., [24,25,27–29,40]). Previous studies have also shown that forecast errors in the ECMWF IFS over Europe were associated with convection over North America [4,12,13] and generally associated with difficulties in accurately representing diabatic processes and their interaction with the wave guide [8,42]. This study also again shows that the forecast error propagates

with the characteristics of a wave packet [34], although the waves in the 20 June case are short wavelength, rapidly propagating Rossby waves. Our finding that a forecast bust and a period of low predictive skill corresponded to high wave number disturbances raises the issue that such high wave number disturbances may have lower inherent predictability. Further investigations of how predictive skill and the occurrence of forecast busts depends on variations in Rossby wave number are needed to address this issue. More detailed investigations into the relationship between the forecast errors and short-comings in the treatment of convection in the ECMWF IFS are also underway.

The error in the forecasts for the two cases studied, as evidenced by the error amplitude and error in the wave activity flux (Figures 7 and 12), was in the ridge downstream of the convective system. This result is also consistent with aspects of the Gray et al. (2014) [43] study that found a decay in total ridge amplitude with lead time that was argued to be consistent with an underestimation of the diabatic enhancement of potential vorticity anomalies. Further evidence of the difficulties in representing these diabatic process in that study was an under-representation of the strength of the weak humidity gradients across the tropopause [43]. Although the general location of the error relative to diabatic heating is similar to our results with the ECMWF IFS, their study was associated with winter season forecasts from three global models.

Our results are also similar to other findings of the role of diabatic heating in interacting with middle latitude flows. A particularly relevant study is the investigation of a predecessor convective event ahead of the extratropical transition of a tropical cyclone over the north Pacific [44] with the precursor convection associated with a ridge, jet streak, and trough. In our study, the diabatic heating amplifies the meridional flow with the trough and ridge with a slowing of the trough. Both events investigated in our study were also associated with an intensification of a jet streak, although in the 20 June case, the intensification of the jet streak was far more pronounced (15 m s$^{-1}$ in the 18 h period (Figure 8). In both cases (11 and 20 June) investigated in our study, the outflow from the convective systems appeared to be closely linked to the intensifying southerly flow associated with the approaching trough. Downstream of this intensifying southerly flow, an area of northerly winds developed to strengthen the ridge. This northerly flow, however, was at a shorter wavelength than the northerly flow with the pre-existing ridge. In the case where the convective systems were not persistent and less rapidly moving (11 June), the shorter-wavelength disturbance eventually dissipated. In contrast, the persistent, rapidly moving MCS associated with 20 June event resulted in the pre-existing longer wavelength disturbance dissipating and the northerlies with the shorter wavelength disturbance strengthening and subsequently moving rapidly downstream.

These findings also suggest the importance of an accurate representation of organized MCSs and properly capturing the interaction of these convective systems with the jet stream dynamics in global models. The observed and predicted precipitation by the ECMWF IFS system is shown in Figure 13. The pattern in the observations (Figure 13a) includes rapidly eastward moving streaks of heavy rainfall at slightly greater 2 day intervals (e.g., beginning on 18, 20, and 22 June) consistent with the rapidly moving, short-wavelength Rossby wave events. Less organized, eastward moving precipitation events occur between these strong convective events with some evidence for a slower movement eastward for these loose envelopes of convective activity (Figure 13a). Tropical Storm Bill is also evident in this figure centered on 97W longitude at on 18 June. This system had a slower eastward movement through this period in the predicted and observed rainfall.

The precipitation forecast produced by the ECMWF IFS is quite accurate in depicting the general rainfall pattern including the rapidly moving streaks of convection, Tropical Storm Bill, and the less organized convection between these events (Figure 13b). A significant difference, however, is that the heavier rainfall predicted in the model tends to be less continuous, especially for the first two streaks of heavy rainfall (first evident on 18 and 20 June). This intermittent rainfall pattern contains continuous streaks of heavy rainfall that repeatedly reform after a short delay resulting in slower movement of the precipitation system as evidenced by the difference in the slope of the observed and predicted precipitation in the Hovmoeller diagram. This slower movement of the precipitation in the model

also occurs in Tropical Storm Bill. Given the possible importance of the persistence of the MCS and its linkage to the jet stream and Rossby wave dynamics, the difficulty in predicting the continuous and rapid movement of the MCS raises the possibility that these subtle shortcomings in the model's parameterization of organized deep convection is also quite relevant to capturing both the coupling of the convection to the Rossby wave activity and an initial source of the error that led to forecast bust for the ECMWF IFS system initialized on the 20 June. The need to capture the divergent outflow at the jet stream level is also likely linked to the upper-level heating. Thus, it is also possible that the amplitude of convective heating profile might be underestimated or the vertical profile poorly prescribed.

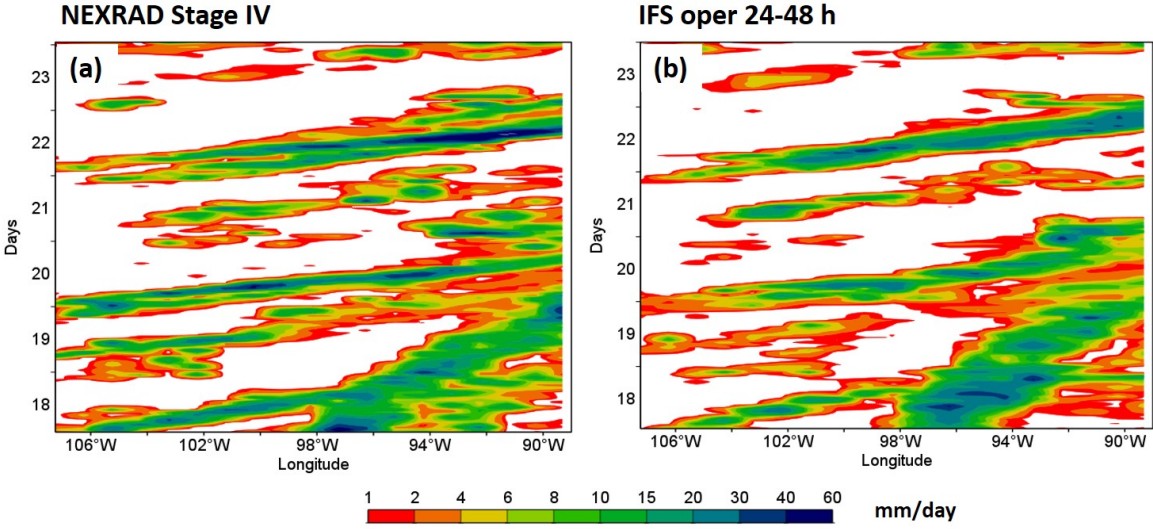

**Figure 13.** Hovmoeller diagrams of hourly rainfall (mm) for the period from 00 UTC on 18 June to 00 UTC on 23 June 2015. The area covered ranges from 33.2 N latitude and −89 longitude to 44.5 N latitude and −107.2 longitude. (**a**) NEXRAD-NCEP (Next Generation Radar—National Center for Environmental Prediction) stage IV rainfall estimates. (**b**) Average of the 24 to 48 h forecast from the ECMWF/IFS.

The ability of the ECMWF IFS to capture the interactions between convection and Rossby wave dynamics was explored by comparing the first guess in the model assimilation system against the observed estimates of rainfall from the operational radar network and the meridional winds at the 200 hPa level from aircraft reports for both the 11 and 20 June cases (Figure 14). Significant differences were found in the rainfall estimates in the first guess fields for both cases with a tendency for displacement errors early in the forecast. The significant errors also occur in the meridional winds in both cases, but with some differences (Figure 14). For example, the differences in the winds for the 11 June case were located both within and outside of the jet stream with a tendency for the first guess to generally exceed the observed meridional winds. In contrast, large differences were more confined to the jet stream in the 20 June case with some tendency for differences of alternating sign. The differences near Tropical Storm Bill to the south of the jet on 20 June were quite small (Figure 14) suggesting that feature was relatively well represented in the model. Unfortunately, the aircraft winds (Figure 14) did not extend far enough north to completely cover the region of large southerly and northerly flow found on the 20 June case (Figure 9) limiting the insight from these comparisons. This finding also implies that relatively few in-situ wind observations are being utilized in the assimilation process and instead the observations in the initial conditions are more likely to be primarily satellite-based with some data from a few radiosonde stations. The extent to which the infrared satellite data in bust cases over the northern Plains is restricted and/or impacted by clouds associated with the MCS is an area for future research.

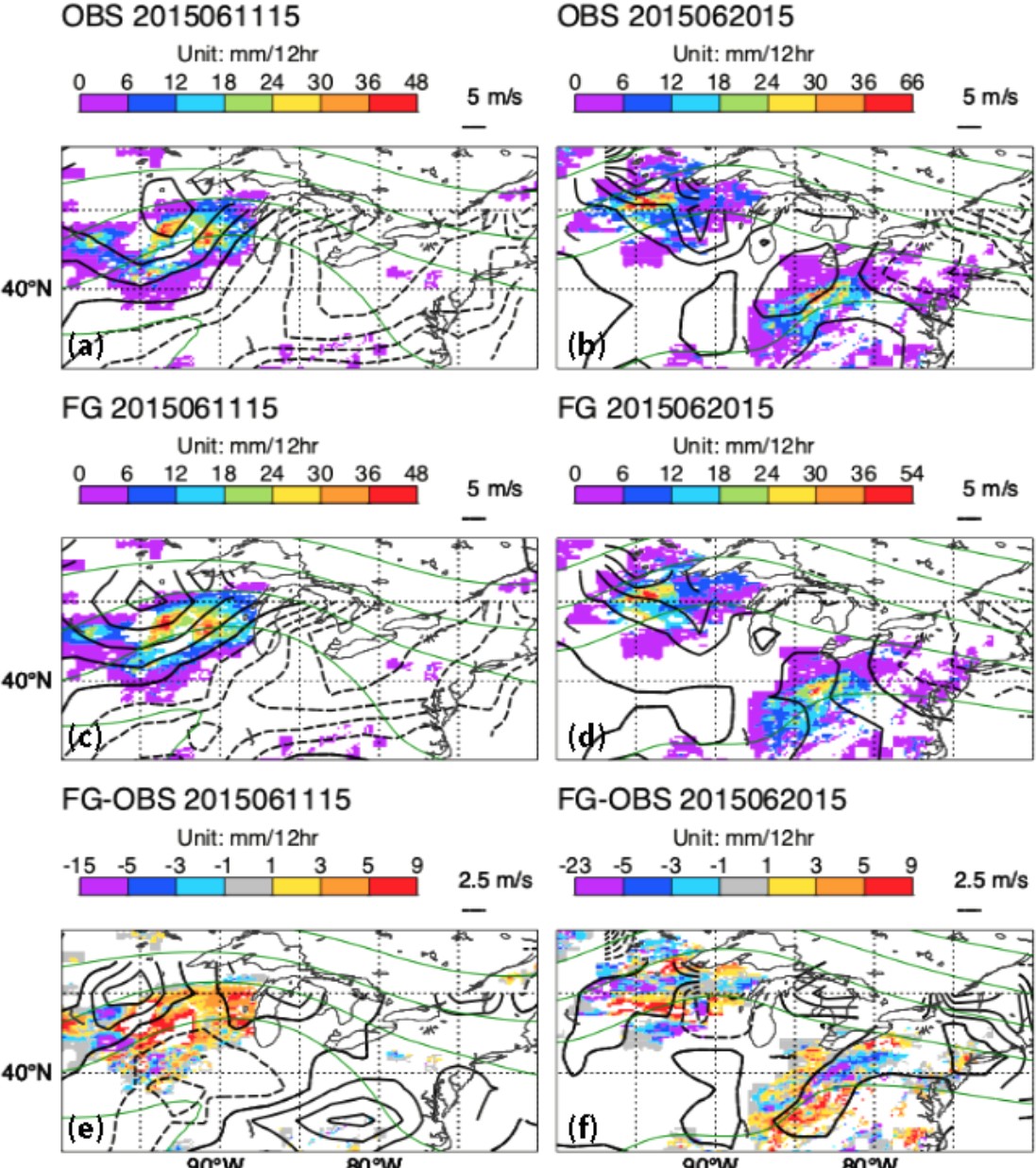

**Figure 14.** Spatial distribution of the 200 hPa streamlines, meridional winds, and the rainfall for the 11 June and 20 June cases. The observations of the meridional winds are from aircraft reports and the observed rainfall estimated from the operational radar network. The first guess of the ECWMF IFS is also shown and all fields are on the grid utilized in the data assimilation system at 15 Local Standard Time (UTC-6) with errors calculated over a 12 h time window of +3 to +15 h. The panels of the left represent the (**a**) observations (OBS), (**c**)first guess (FG), and (**e**) difference fields (FG-OBS) for 11 June, while panels on the right (**b**), (**d**) and (**f**) represent the same fields, but for 20 June.

*4.2. Representation of the Diurnal Propagation of Deep Convection*

The linkage between organized convection over the Great Plains and forecast errors downstream over Europe raises the specter of how well operational models represent nocturnal convective systems. This issue is important given the long known tendency for the region from the central and northern Plains to near the Great Lakes [45] to be characterized by an nocturnal maximum in deep convection over the summer. Representing these nocturnal convective systems is a challenge given the need to accurately depict the characteristics of the component of the nocturnal flow including the stable

boundary layer and the low-level jet [17]. An additional challenge is that the interaction between the convective outflows from these nocturnal MCSs and the ambient flow tends to lie in dynamic regime that will produce bores [31,46]. These bores, in turn, produce substantial lifting (between 500 m and over 1 km) in the lower troposphere [47,48] that destabilizes the large portions of environment surrounding the convective system [48].

The diurnal variations in warm season, deep convection over the central U.S. has been addressed in numerous studies showing that this nocturnal convective activity begins over the elevated portion of the continent over and near the Rocky Mountains during the late afternoon and early evening and subsequently grows up-scale and propagates eastward as a loose envelope of deep convection that passes over the Great Plains during the night. The Carbone and Tuttle (2008) study [49] is particularly relevant as the study covered an extensive period from 1996 to 2007. The Hovmoeller diagram for rainfall from 11 to 15 June is shown in Figure 15. The observed rainfall pattern reveals a precipitation structure consistent with this eastward propagating envelope of convection (Figure 15a) with precipitation beginning over the western edge of the domain and moving eastward. The eastward movement of the rainfall in the observations has a similar slope in the Hovmoeller diagram as the 17 (m s$^{-1}$) movement found in the Carbone and Tuttle climatological study [49] (Figure 15).

The rainfall pattern predicted by the ECMWF IFS (Figure 15b) tends to capture the observed eastward progression, but accurately representing the location and slope of the rainfall in the Hovmoeller diagram appears to be difficult for the model. The model also appears to under represent the frequency of the heavy rainfall events associated with these eastward moving envelopes of deep convection. The result that representing the rainfall pattern during this 11–15 June period is more difficult to represent in the forecast model than the rapidly moving MCSs during the 18–22 June period (Figure 13 is somewhat expected as the convection in the later period is associated with stronger synoptic forcing. These short comings in the treatment of deep convection in the model during the 11–15 June period do not, however, result in forecast busts. The larger and more rapid error growth in the 18–22 June period when the precipitation appears to be more well represented in the model points to the demanding nature of accurately portraying the interaction between the MCS and the large-scale as the significant error forms within the jet stream and subsequently grows as a Rossby wave within the jet stream as a wave guide.

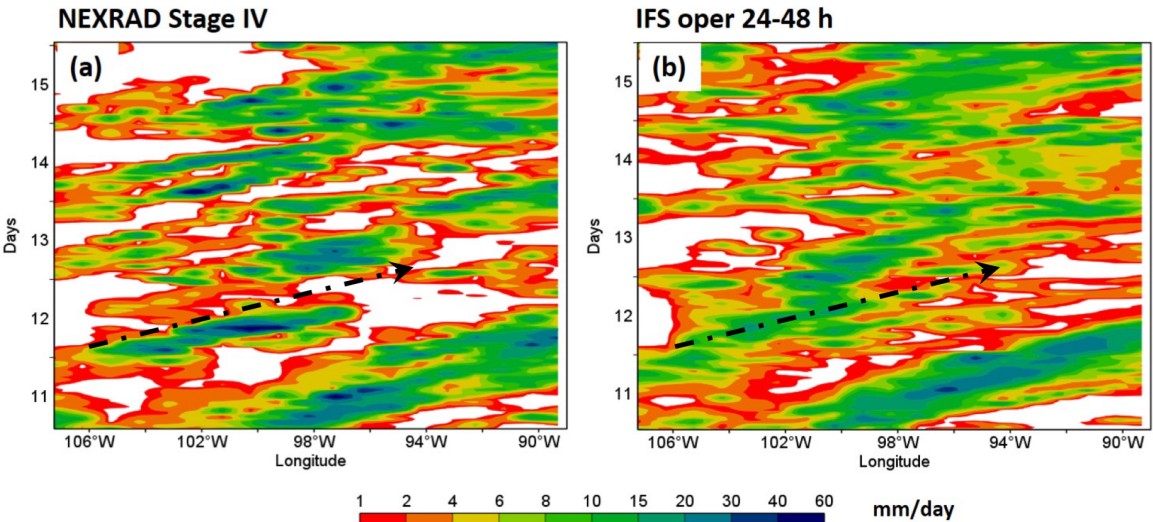

**Figure 15.** Hovmoeller diagrams of hourly rainfall (mm) for the period from 00 UTC on 11 June to 00 UTC on 16 June 2015. The area covered ranges from 33.2 N latitude and −89 longitude to 44.5 N latitude and −107.2 longitude. (**a**) NEXRAD-NCEP stage IV rainfall estimates with the dot-dash arrow indicating convection moving eastward at 17 (m s$^{-1}$) as found in [49]. (**b**) Average of the 24 to 48 h forecast from the ECMWF/IFS.

### 4.3. Implications for Convection over North America and Forecast Skill Downstream

The MCSs investigated suggest the appearance of "coupling" of the convective systems with Rossby wave dynamics as the upper-level outflow from long-lived, MCSs near and south of the jet stream seem to enhance southerly flow ahead of the trough in the Rossby waves (Figures 6 and 9). Subsequently, the northerlies intensified downstream strengthening the ridge at a relatively short wavelength with a more sustained impact in the 20–21 June system. The strengthening of the southerly flow does not suggest a symmetric outflow from the MCS, with both northerly and southerly anomalies, such as might be expected in tropical cyclone. Rather the response is consistent with an outflow associated with intense rear-to-front flow with a southerly component as was noted in simulations of an MCS on 11–12 June [39].

The cyclonic low pressure system and the pattern of high CAPE in the 20–21 June MCS (Figure 10) is further evidence of the coupling between the MCS and the Rossby wave. This MCS and accompanying low pressure system initially formed near a stationary front. The 11 June MCS was also associated with a similar evolution. The observed surface analysis for 12 UTC on 20 June (Figure 16a) also shows that surface conditions include a cold front, a weakening warm front, and a low pressure system. An extensive area of heavy precipitation is located near and primarily ahead of the low pressure. This operational surface analysis also includes a trough extending north of the surface pressure system. The accompanying 24 h forecast from the ECMWF (Figure 16) shows the difficulty that the model forecast has representing this low pressure system with the predicted low located to the southwest of the observed feature associated with the heavy rainfall. In contrast, Tropical Storm Bill (see closed 1008 hPa contour over the Ohio River Valley) is relatively well predicted with smaller errors. However, both low pressure systems in the model do appear to move slightly slower than the observed feature.

The observed development of a surface low near the MCS and the close linkage between the MCS outflow and the Rossby wave dynamics at the jet stream aloft is reminiscent of a diabatic Rossby wave (DRW). An excellent review of these features is captured in the climatological study of Boettcher and Wernli (2013) [50]. Their study also discusses how the terms DRW and diabatic Rossby vortex often refer to the same phenomena. DRWs have drawn significant attention in terms of rapidly deepening, strong cyclones over the North Atlantic, such as, the 1991 "perfect storm" [51], the damaging Lothar cyclone in 1999 [52,53], the 24–25 February 2008 east coast snow storm [54], and the rapidly intensifying December 2005 storm [55]. Our study raises the possibility that over continental locations, such as the Great Plains, the triggering and maintenance of long-lived MCS close to the jet stream and surface baroclinic zones can lead to surface cyclogenesis, enhanced diabatic heating, and an intensification of the surface cyclone, such as can occur with a DRW. Previous work [50] has noted that the remnants of an MCS moving over a baroclinic zone is one mode of generating a DRW. We also note that the early explorations relevant to DRWs were associated with idealized simulations such as organized convection in baroclinic zones or regions of high vertical shear [56–58].

Our study with the close linkage of MCS to southerly flow in the jet stream associated with Rossby waves, the surface cyclonic development along the front and the response downstream on the wave guide with the development/intensification of a Rossby wave and the enhanced jet streak raise the possibility that the framework of DRWs may be relevant to certain MCS. This concept is consistent with aspects of studies in the past literature. For example, Maddox (1983) [59] investigated the synoptic situation associated with several large mesoscale convective complexes (MCCs) finding that the MCCs: (i) Formed near a stationary, surface front and ahead of the short-wave trough aloft; (ii) became closely linked to the eastward propagation of the trough; (iii) an intensification of the trough occurred late in the lifetime of the storm system. Other studies have found MCCs to take on an open wave low pressure and frontal structure [60,61]. In addition, other MCSs, such as derechoes, have also been shown to be associated with convection developing within a baroclinic zone, typically associated with a quasi-stationary zone oriented perpendicular to a convectively unstable flow with a cyclonic, low pressure systems developing in some cases [62]. Another possible way that some MCSs behave similar to DRWs is the ascent that occurs due to frontal circulations that favorable preconditioning of the

environment ahead of the front. Thus frontal ascent can maintain the MCS as has been discussed in the literature [63,64], which are highly relevant to the observed flow evolution in our study.

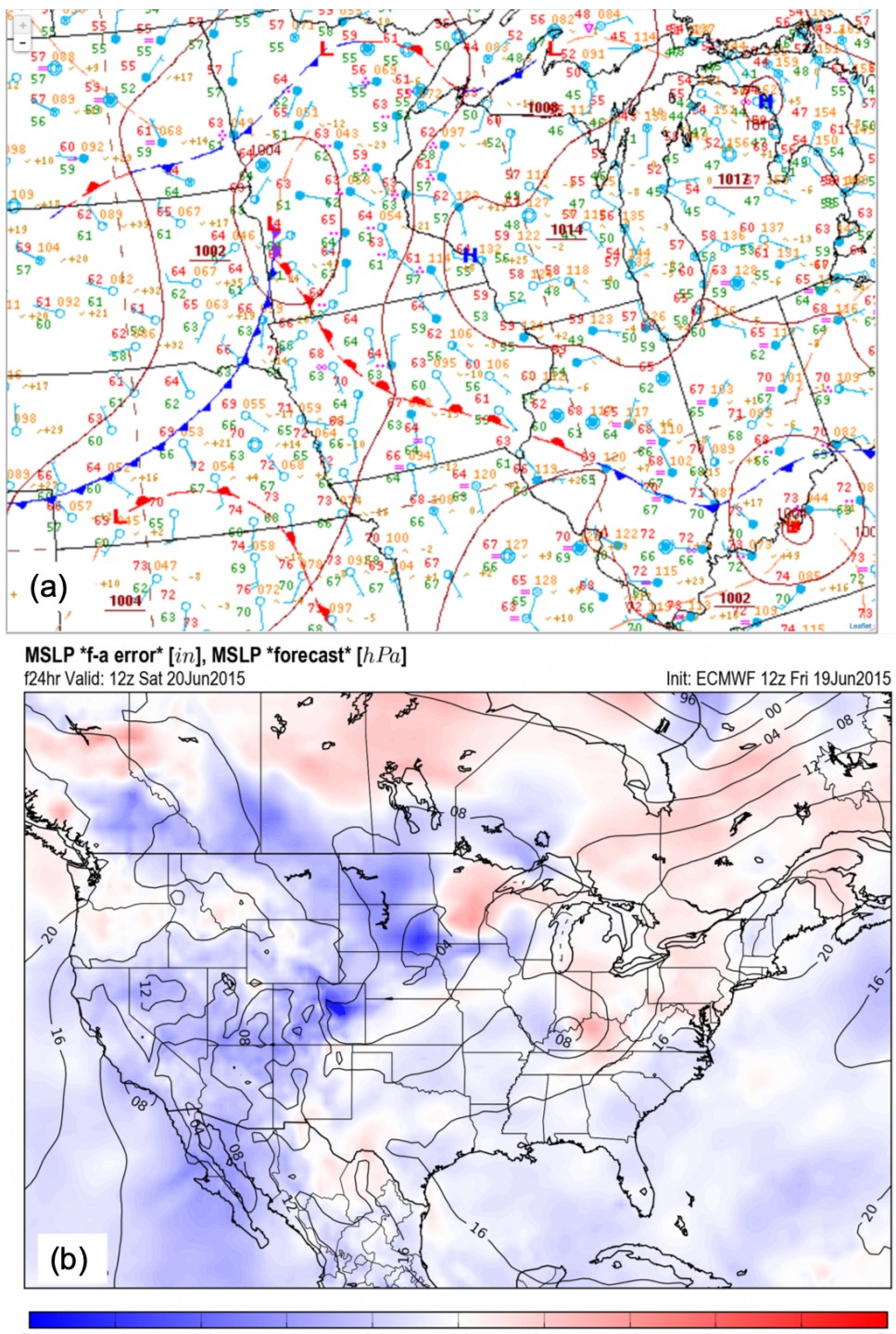

**Figure 16.** Characteristics of the surface weather for 12 UTC on 29 June. (**a**) Operational surface analysis from the NOAA's National Weather Service's Weather Prediction Center (NWS/WPC) showing surface station plots in standard format, sea level pressure (hPa - 1000), temperature, dew point temperature, and rainfall along with a surface pressure and frontal analysis (standard symbols). (**b**) The 24 h forecast for 12 UTC on 20 June by the ECMWF IFS initialized at 1200 UTC on 20 June 2015. The contours represent the predicted surface pressure (mb) and the color coding represents the error in surface pressure (forecast - observed.)

While these points raise the issue of linkage between Rossby waves and warm season convection, the extent to which the concept of DRWs may be a relevant concept for MCCs and other MCSs over this region, the surface cyclonic circulations are not particularly dramatic and, hence, the extent of the coupling may likely be relatively weak. Still these results suggest that convection during bust events is strong enough to be linked to and impact jet stream dynamics (i.e., synoptically forced). The subsequent development of the surface low and the advection of a high CAPE air mass suggests a positive impact of the large-scale response on the MCS. The linkages between severe weather over North America and forecast errors in the medium-range over Europe was noted earlier by Rodwell et al. [12] study of uncertainty growth in the ECMWF ensemble system. The case investigated for that study had 631 high wind, 194 hail, and 89 tornado events over the continental U.S. recorded in the official severe weather reports from the NOAA Storm Prediction Center. The 9–11 April 2011 case studied in the earlier work of Rodwell et al. was also associated with a major severe weather outbreak as 917 severe weather reports were issued during this period. For 21 and 22 June 2015 (Figure 17) case in our study, severe weather was again quite extensive with well over 700 reports that were primarily related to wind (548 reports with 18 over 65 knots), but included numerous tornadoes (33 reports) and hail (152 with 29 larger than 2" in diameter). The general spatial pattern of the reports is consistent with a pattern of rapidly eastward moving systems consistent with the short wavelength Rossby wave packets (Figure 11). News reports generally noted the similarity of the radar images and storm damage on 22 June to a derecho and noted that the impact included 500 flights canceled from Chicago alone. The characteristics of the storm damage suggest that the case meets the classic definition of a derecho event [62] with the radar observations taking on the general bow shaped criteria suggested by [65].

The apparent linkage between widespread severe weather over the Great Plains and forecast busts over Europe in the three bust cases shown (this study and the two events investigated in the previous Rodwell et al. studies) suggest significant societal benefits to both Europe and North American will result from improvements in forecast systems designed to reduce the frequency of forecast busts. The correspondence between forecast busts over Europe and the tendency for severe weather in the initial conditions and early forecast evolution over North America deserves future attention.

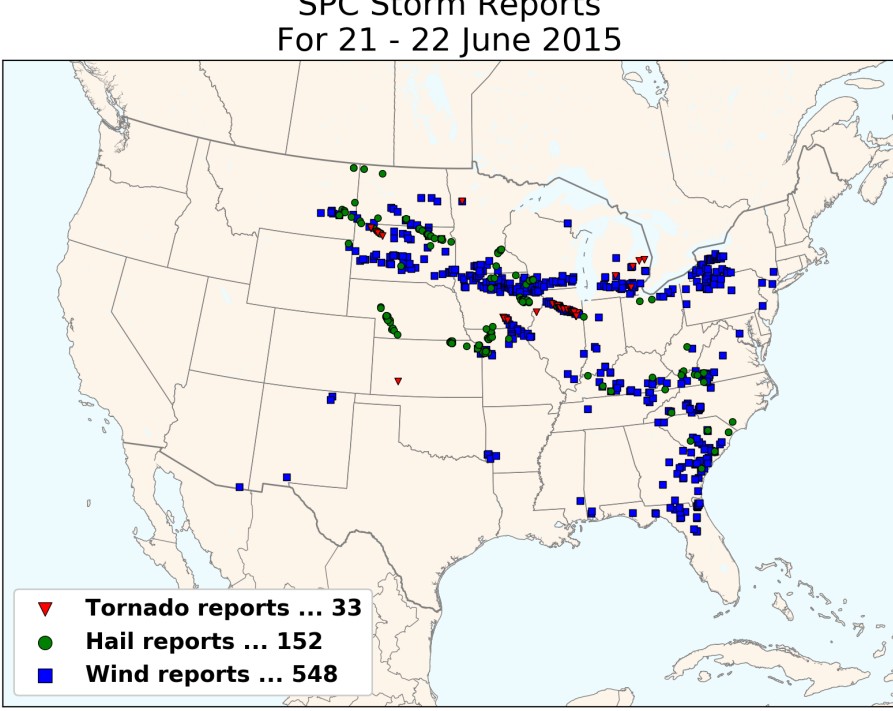

**Figure 17.** Reports of severe weather from NOAA/NWS/Storm Prediction Center for 21 and 22 June 2015.

### 4.4. Downstream Error Propagation Including Linkages to Arctic Circulations

The period with lower forecast skill over Europe (Figure 1) including the bust on 20 June was associated with strong zonal flow (Figure 2) over the central Plains. This strong jet was also characterized by interaction with flow from the Arctic. Specifically, the strong winds in the middle latitude jet over central North America coincided with the merger of strong flow out of and back towards the Arctic (Figures 4 and 8). One implication of this flow regime is that errors associated with difficulties in representing MCS over North America in the model can propagate into the Arctic. This behavior can be seen in the ECMWF IFS forecast initialized on 12 UTC on 21 June again utilizing the error amplitude and error in the wave activity flux (Figure 18). At 24 h into the forecast (Figure 18a), the error amplitude once again shows the initial error in the ridge consistent with the other convective events shown and past work [43]. The error subsequently appears in the wave activity flux at 36 h (Figure 18b) again showing that the error induced by convection has now taken on the form of a Rossby wave packet. At 48 h into the forecast (Figure 18c), the error amplitude is largest within the trough consistent with the rapid downstream propagation of the wave packet. At this time, however, large error amplitude and error in the wave activity flux also appear to be propagating toward the Arctic. This propagation into the Arctic is more clearly evident at 60 h into the forecast with errors in the wave activity flux and large error amplitude in the vicinity of Greenland (Figure 18d). At this time in the forecast, the large error amplitude also continues to propagate rapidly across the North Atlantic toward Europe.

The propagation and amplification of the error continues in the Arctic and mid-latitude as evident in the 72 and 96 h forecasts (Figure 18) producing a large area of significant errors at 96 h across Europe and large expanses of the Arctic from Greenland to over and to the north of Siberia. The poleward orientation of the vectors indicating the error in the wave activity flux suggest that these errors are associated with anticyclonic wave breaking. The propagation of middle latitude errors originating over the central Plains due to convection into the Arctic is expected from the systematic studies of forecast busts of Lillo and Parsons (2017) [8] and the results of the composite study of Rodwell et al. (2013) [4] that show regime changes across the Arctic occur with these busts.

The possibility that errors in middle latitude forecasts are introduced into the Arctic and subsequently expand and propagate across the Arctic as a Rossby wave deserves a more systematic investigation. Some additional evidence for this process is that the error in the wave activity flux in the 96 h forecast for 20 June (Figure 12d), which shows errors in the nature of Rossby waves over the Arctic Ocean north of Siberia and the start of wave breaking and error amplitudes over Greenland. However, the finding that the errors in the Arctic propagates as a Rossby wave packet is somewhat unexpected given that vortices rather than Rossby waves tend to dominate in the Arctic given the higher planetary rotation rates [66]. The question of whether the error physically corresponds to Rossby wave packets moving through the Arctic or simply arises as an artifact of the difference between the forecast and the observed flow is an area for future investigation. The regime transition in the Arctic associated with forecast busts [4,8] suggests the occurrence of physical changes in the flow. Another Arctic-middle latitude interaction to explore is the possibility that the merger of these two strong flows over North America, one originating over the Pacific and the other out of the Arctic, played a role in supporting these unique short-wavelength Rossby waves with rapid propagation.

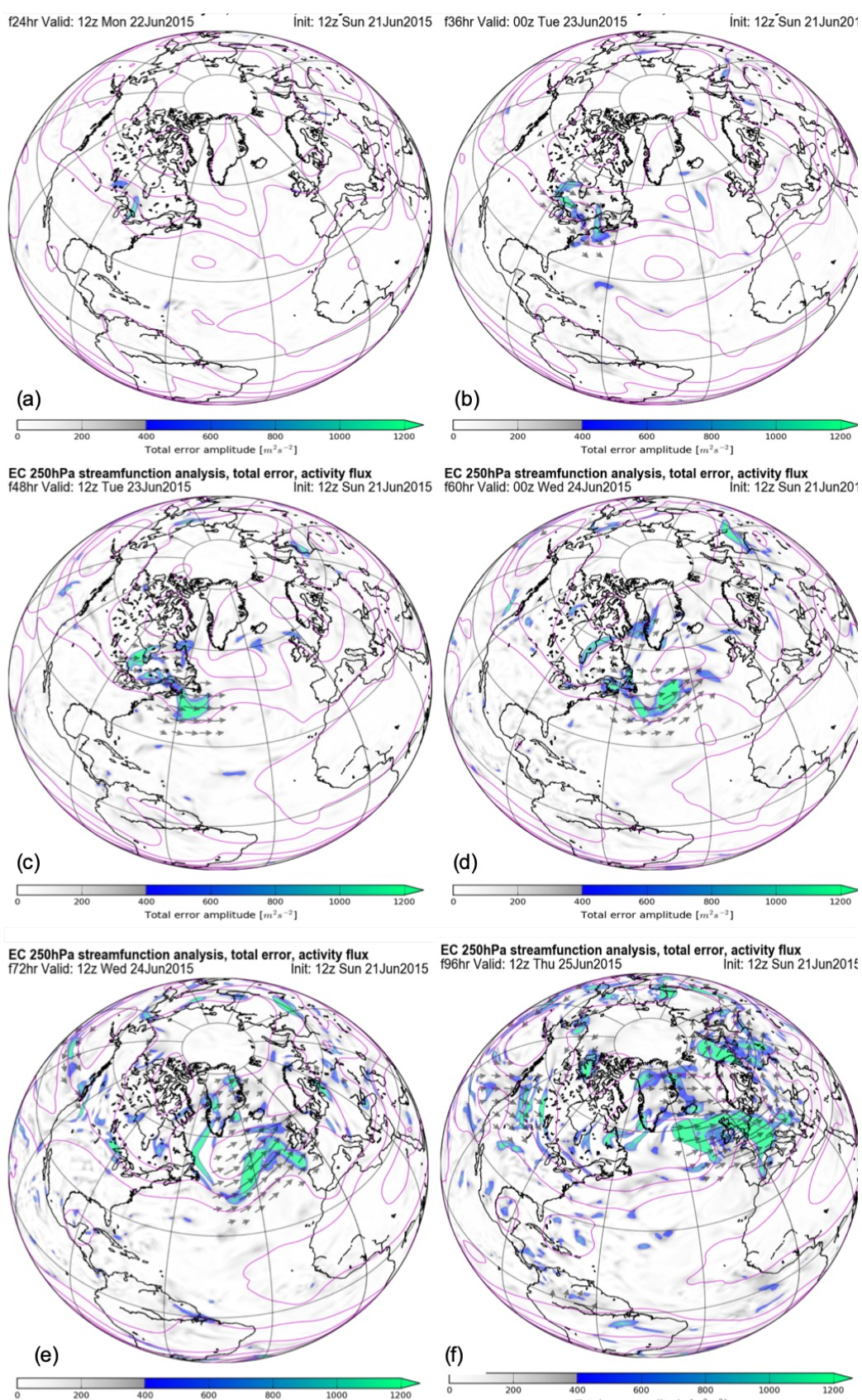

**Figure 18.** Streamlines, error amplitude, and error in the wave activity flux for the ECMWF IFS forecast initialized at 12 UTC on 21 June 2015. Forecast at (**a**) 24 h, (**b**) 36 h, (**c**) 48 h, (**d**) 60 h, (**e**) 72 h, and (**f**) 96 h.

## 5. Conclusions

The error initiation and growth in middle latitudes (Figure 18 and the other cases presented) suggests the following framework. First, errors in the forecast are initially linked to problems in the representation of MCSs sufficiently near the jet stream. The errors appear near the ridge in the Rossby wave with a characteristic arc-shape in the error amplitude during the first 12 to 24 h into the forecast in these cases, although one suspects that this error in convection could happen at any time during the forecast. This early error initiation and growth is similar to the first stage of error proposed by Zhang et al. (2007) [26]. While their study discussed the potential importance of gravity waves, our work suggests that the enhanced southerlies were likely the result of strong outflows associated with front-to-rear flow. Another difference is that while [26] propose that the impact of convection in error growth is relatively short-lived of 6 h, the convective events in this study lasted for on the order of 1 day with coupling to the large-scale meridional motions. The long lasting MCSs in our study with enhanced southerly flow associated with the trough imply that if the evolution and longevity of the MCS in the model are not well represented, errors may arise in the model treatment of Rossby wave dynamics. The importance of a linkage between MCS and upper-level Rossby wave dynamics is consistent with the recent finding of Weckwerth and Romatchke (2019) [67] that found that the top 10% of rain-producing events during the summer 2015 period that included PECAN were linked to an approaching trough. While our study suggests a coupling of the outflow and the Rossby wave, further investigations are underway to determine the nature of the error in the model (e.g., errors in the location and timing of convective initiation, gravity waves generated by MCSs, inability of the model to maintain nocturnal convection, or inaccuracies in the vertical profile of heating and divergence).

The second stage in the Zhang et al. (2007) [26] study was associated with a transition of the errors to the larger-scale introducing errors in the balanced flow. The cases investigated herein were associated with deep convection over the Great Plains near the jet stream. In these events, the varying degrees of persistence in the MCSs was difficult to represent in the ECMWF IFS system, whether the MCSs were rapidly propagating in an unusual regime where short-wave length Rossby waves were present or part of the loose envelope of eastward propagating convection that forms over the higher elevations in association with diurnal heating. The southerly meridional flow ahead of the trough in the Rossby wave packet was intensified near the location where convective outflows would occur in MCSs with strong front-to-rear inflow. The impact of convection included the intensification of a jet streak and downstream northerlies associated with a strengthening ridge. These changes suggest that the geostrophic adjustment process is responsible for the strengthening of the jet streak and the error being associated with the modification or triggering of a Rossby wave packet.

The third stage proposed by Zhang et al. (2007) [26] is when the errors grow with large-scale baroclinic instability. Their study notes that this growth depends on the nature of the waves. This downstream growth was noted in our study as the errors propagated downstream and amplified resulting in decreased forecast skill over Europe. In general, this later growth was not a focus of our study. However, our study included a bust and poor forecasts in a period with large (7 to 13) wavenumber disturbances with rapid propagation. In this regard, we note the previous findings [30] that these large wavenumber disturbances have large error growth rates.

While our analysis utilizing the error amplitude and error in the wave activity flux suggests that one area of investigation is the treatment of convection, errors in the initial conditions may also play a significant role in forecast busts [4,68–70]. These errors in the initial conditions could play a role in the difficulty in representing MCS, although the difficulty in accurately representing the persistence of both the rapidly propagating MCS and the general eastward progression of the envelop of deep convection across the continent east of the Rockies [49] also appear to be important areas for future work. The researchers involved in this study are undertaking more detailed comparison between the NEXRAD radar data and the model treatment of deep convection during this period in an effort to understand the shortcomings in the parameterization of deep convection.

Another area for future work is to advance understanding of the tendency for MCSs to drive shorter wave length disturbances in both flow regimes studied as evidenced by the location of where northerly winds were generated on the jet stream. This result, together with previous investigation [4,71–73] suggest the need for further studies into how the scale and persistence of middle latitude MCS influence Rossby wave dynamics. For example, Stensrud (1996) [71] argue that individual MCSs with lifetimes of 6 to 18 h are not sufficient to influence the large-scale environment, but the cumulative impacts of persistent convection from multiple MCSs can act as a Rossby wave source and modify the environment over 50 degrees longitude. The individual MCS also modify the inflow to favor the continuation and persistence of deep convection. In contrast, this study and the hypothesis of Rodwell et al. (2013) suggests that a strong, propagating MCSs near the jet stream can intensify or trigger Rossby wave disturbances initially through the amplification of the ridge. The forcing from these MCSs in the 20–21 June case tended to drive the atmosphere toward shorter wave length disturbances. The issue of how the wave response depends on the nature of the jet stream itself and the magnification of forecast errors generated over North America through interaction of the diabatic heating associated with the warm conveyor over the Atlantic are other areas for future work.

The Rodwell et al. case studies [4,12] and this investigation were associated with large outbreaks of severe weather. These bust cases were not selected on the basis of high impact weather over the Great Plains, but rather the occurrence of large forecast busts. The occurrence of large, widespread outbreaks of severe weather in association with forecast busts raises the possibility that the intensity of convection also plays a role in generating the initial errors that lead to forecast busts. Finally, the relationship of Rossby wave responses induced by convection over the Great Plains to circulation changes in the Arctic, as in the 20 June 12 UTC forecast and previous work [4], and the frequency with which Rossby waves propagation across the Arctic cause regime change and the loss of forecast skill [8] in the Arctic is also an area for future work. The linkage between severe storms over the Plains and forecast errors and regime changes in the Arctic underscores the global, interconnected nature of our atmosphere.

Finally, we note that since the excellent pioneering studies of Lorenz [18] researchers involved in predictability studies have often framed the impact of small-scale disturbances on the large-scale flow into questions of whether weather events and forecast failures at far distances could be induced by the flap of a sea gull's or a butterfly's wings. Our study reinforces the idea that small-scale disturbances, such as convective systems, are indeed important to generating disturbances at far distances. However, using this terminology, our results also suggest that the location of the flapping (near the jet stream) is indeed critical to the initiation and potential for subsequent growth of errors. The intensity and duration of the flapping is also important given the sustained interaction that was observed between long-lived, severe MCSs and Rossby wave dynamics. The high frequency of intense MCSs over North America and their correlation with initial forecast errors and the subsequent error growth in the jet stream over the North Atlantic raises the possibility that the loss of predictability may systematically vary with location. Hence, our findings basically reinforce Lorenz's caution mentioned in the introduction that predictability studies should consider that the location of those small-scale disturbances that can disrupt the flow is non-random relative to synoptic-scale features.

**Author Contributions:** For conceptualization, D.B.P., S.P.L., M.J.R., and P.B.; methodology, S.P.L., D.B.P., P.B., C.P.R., and C.M.B.; software, S.P.L., C.P.R., P.B., and C.M.B.; validation, S.P.L., P.B., and C.M.B.; formal analysis, S.P.L., D.B.P., P.B., and C.P.R.; investigation, all; resources, D.B.P.; data curation, P.B., S.P.L., and C.P.R.; writing—original draft preparation, D.B.P.; writing—review and editing, P.B., S.P.L., and M.J.R.; visualization, S.P.L., P.B., and C.P.R.; supervision, D.B.P.; project administration, D.B.P. and P.B.; funding acquisition, D.B.P.

**Funding:** In terms of external funding, this research was funded by the U.S. Office of Naval Research grant entitled "Understanding and Improving Prediction of the Impacts of Rossby Wave Breaking on Tropopause Polar Vortices and Arctic Cyclones", which is ONR grant number N00014-18-1-2163. The involvement of Mr. Connor Bruce was supported by funding from the University of Oklahoma's Undergraduate Research Opportunities Program (UROP).

**Acknowledgments:** We wish to acknowledge the European Centre for Medium Range Weather Forecasts (ECMWF) for supporting this effort and note that the collaboration was initially encouraged by Alan Thorpe, former Director General of ECMWF. The lead author also appreciates scientific discussions on predictability with Peter Bauer

and Florence Rabier, the ECMWF's current Director General. We also appreciate thoughtful discussions with Steven Cavallo and Ben Schenkel (both of the University of Oklahoma) and colleagues involved in the Office of Naval Research team on Arctic cyclones including James Doyle (Naval Research Laboratory) and Lance Bosart (State University of New York at Albany). In addition, the paper was improved in response to the suggestions and comments of the editors and the three anonymous reviewers. We also acknowledge the University of Oklahoma's First Year Research Experience (FYRE) program that resulted in the involvement of Connor Bruce. Finally, we note with sadness the passing of Fuqing Zhang of Penn. State U. who was an enthusiastic leader in this area of research.

**Conflicts of Interest:** The authors declare no conflict of interest.

**Abbreviations**

| | |
|---|---|
| MDPI | Multidisciplinary Digital Publishing Institute |
| DOAJ | Directory of open access journals |
| ACC | Anomaly Correlation Coefficient |
| CAPE | Convective Available Potential Energy |
| DRW | Diabatic Rossby Wave |
| ERA | ECMWF Reanalysis |
| ECMWF | European Centre for Medium Range Weather Forecasts |
| IFS | Integrated Forecast System |
| MCS | Mesocale Convective System |
| NAO | North Atlantic Oscillation |
| NWP | Numerical Weather Prediction |
| NAWDEX | North Atlantic Waveguide and Downstream Impact Experiment |
| PECAN | Plains Elevated Convection at Night |
| PNA | Pacific-North American |
| RMSE | Root Mean Square Error |
| TIGGE | THORPEX Integrated Grand Global Ensemble |
| UTC | Coordinated Universal Time |

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
