# Peer review of "The Role of Continental Mesoscale Convective Systems in Forecast Busts within Global Weather Prediction Systems"

_atmosphere, doi:10.3390/atmos10110681_

Round 1

Reviewer 1 Report

This manuscript provides an extensive literature review of the influence of convective systems on the predictability in global NWP models. In addition, the manuscript adds to the body of knowledge with the study of forecast busts associated with MCSs over the central plains of the US in June 2015. The paper is well written and well organized, and I recommend it be published pending minor review. Please see comments below.

List of authors. What is the meaning of the cross superscript after Parsons’ name? Lines 13-14 and 430. The authors claim that improvements are needed to the parameterization of organized deep convection in NWP models. To my knowledge, models do not have parameterization of organized convection, they simply have deep convection parameterizations. It is not clear that a parameterization of organized deep convection is needed, because the operational global models, such as the ECMWF IFS used in this study, should be able to resolve MCSs. Unless the authors can explain this further, I recommend changing the sentence to speak of the need for improvements in the representation of organized deep convection in NWP models. Line 64. Other factors (...) are. Lines 102 and 109. Zhang et al. Line 194.  Pattern (...) undergoes Line 205. June is characterized by a wave numbers Line 212. Add just before below (because 0.1 is also below 0.3). Line 278. Examination (...) reveals Line 280. 06 UTC is not shown in the figure. Line 375. Will be discussed. Line 387.  20-21. Line 450. Shortcomings Line 482. Explored by. Line 486. It is not clear to me that the model underrepresents rainfall because in Fig. 16 I see a lot of orange shades. Please clarify. Line 610. Three? The paragraph starting on Line 141 says that only one bust occurs in this period. Line 642. Expand and propagate. Line 648. Error is physically corresponds. Lines 653 and 714. America. Line 657. Improve the sentence: errors are initially linked to errors. Line 660. Time (not tine). Line 664. growth is relatively short-lived (6 h) (not growth us relatively short-lived of 6-h). Line 666. The long lived MCSs (...) imply. Line 667. evolution and longevity (...) are. Line 675. the convective systems showed. Lines 677, 701, 705, 707, 708, 712, 734. Use MCSs instead of MCS. Line 692. Analysis of the error. Line 716. The word case does not fit well in this sentence. Line 722. Frequency with which Line 736-737. that the timing of loss of predictability may instead of that when predictability is lost may. Line 754. Dr. Florence Rabier. Acronyms. ACC and RMSE are defined twice. The list of abbreviations on page 38 is short and many additional abbreviations are used in the manuscript NAO, PNS, MCS, etc.). Figure 2. Coefficient (not co-efficient). Figure 6. 06 UTC (not 96 UTC). Figure 7. Replace with a higher-resolution figure. Mention in caption that red/blue contours are positive/negative. The time period is (...) to 04 UTC. Figure 10. Replace with a higher-resolution figure. Mention in caption that red/blue contours are positive/negative. Figure 13. Caption is missing c) and d). Figures are missing a), b), c), d) labels. Figure 14. Add units. Figure 16. and the rainfall for both the. Map is hard to see. Is this just one time zone (otherwise the concept of local time does not apply)? How can you match an analysis with observations over a 12h hour period with an instantaneous observation? Figure 18. 12 UTC on 19 June. (mb - 1000). Figure 20. Caption is missing c) and d). Figures are missing a), b), c), d) labels. Figure 21. Consolidate with Figure 20 since this is just two additional forecast lead times. Figures 20-21. According to the first paragraph on Page 4 and Figure 2, the definition of forecast busts used in this study pertains to the 6-day forecast over Europe.  In that case, why not show the 6-day forecast in Figures 20-21 to complete the demonstration of the source of errors for the 6-day forecast? References. Sometimes journal articles are in lower case, other times in upper case, e.g., Monthly weather review and Monthly Weather Review.

Author Response

The authors thank the reviewers for their careful reading of the paper, thier recommendation of minor revisions, and their suggestions for improvement. We made the changes suggested by this reviewer noting that the suggestions were often typos, grammatical errors, and clarifications in wording.

Please see attachment noting one difference from the attachment is that we did combine the last two figures into one but could not extend the plots in the figure to 144 h.  Just too small.

Reviewer 2 Report

See attached pdf file

Author Response

Please see the attachment. We replied to all the comments and revised the manuscript. We shortened the paper and reduced the number of figures by three. We did not, however, do the proposed changes to the introduction as the other two reviewers commented that the paper was well written and had positive, favorable comments on the extensive literature.

Author Response

Please see attachment. We note the reviewer's recommendation was minor revisions in their text. However, we took the reviewer's comments seriously and made small changes/corrections in the manuscript to reflect their suggestions.

Round 2

Reviewer 2 Report

I have no more comments